# Arctique: An artificial histopathological dataset unifying realism and controllability for uncertainty quantification

**Jannik Franzen**[1,2,6,*,✉], **Claudia Winklmayr**[1,*], **Vanessa E. Guarino**[1,5,*], **Christoph Karg**[1,*],
**Xiaoyan Yu**[1,5], **Nora Koreuber**[1], **Jan P. Albrecht**[1,5],
**Philip Bischoff**[2,3,4], **Dagmar Kainmueller**[1,6,✉]

[1] Max-Delbrück-Center for Molecular Medicine in the Helmholtz Association
and Helmholtz Imaging, Berlin, Germany
[2] Charité—Universitätsmedizin Berlin, Corporate Member of Freie Universität Berlin
and Humboldt-Universität zu Berlin, Institute of Pathology, Berlin, Germany
[3] Berlin Institute of Health at Charité—Universitätsmedizin Berlin
[4] German Cancer Consortium, German Cancer Research Center, Partner Site Berlin
[5] Humboldt-Universität zu Berlin, Faculty of Mathematics and Natural Sciences, Berlin, Germany
[6] Digital Engineering Faculty of the University of Potsdam

✉ {firstname.lastname}@mdc-berlin.de    * equal contribution

## Abstract

Uncertainty Quantification (UQ) is crucial for reliable image segmentation. Yet, while the field sees continual development of novel methods, a lack of agreed-upon benchmarks limits their systematic comparison and evaluation: Current UQ methods are typically tested either on overly simplistic toy datasets or on complex real-world datasets that do not allow to discern true uncertainty. To unify both controllability and complexity, we introduce Arctique, a procedurally generated dataset modeled after histopathological colon images. We chose histopathological images for two reasons: 1) their complexity in terms of intricate object structures and highly variable appearance, which yields challenging segmentation problems, and 2) their broad prevalence for medical diagnosis and respective relevance of high-quality UQ. To generate Arctique, we established a Blender-based framework for 3D scene creation with intrinsic noise manipulation. Arctique contains up to 50,000 rendered images with precise masks as well as noisy label simulations. We show that by independently controlling the uncertainty in both images and labels, we can effectively study the performance of several commonly used UQ methods. Hence, Arctique serves as a critical resource for benchmarking and advancing UQ techniques and other methodologies in complex, multi-object environments, bridging the gap between realism and controllability. All code is publicly available, allowing re-creation and controlled manipulations of our shipped images as well as creation and rendering of new scenes.

## 1 Introduction

The crucial importance of reliable UQ for the deployment of segmentation algorithms to safety-critical real-world settings has long been recognized by the machine learning community, and the field has seen substantial development of methodology over past years (see e.g. [12, 1, 31]). However, there is a glaring lack of comprehensive evaluation of UQ methods, which makes it difficult to contextualize new methods within the existing paradigms, and renders the choice of suitable UQ

38th Conference on Neural Information Processing Systems (NeurIPS 2024) Track on Datasets and Benchmarks.

methods burdensome for practitioners. One reason for the lack of comparative insight is that often UQ methods are developed from theoretical considerations and tested on hand-crafted toy datasets but fail to provide meaningful, interpretable results on complex real-world datasets [20, 6].

Towards more insightful benchmarking of UQ methods, it is desirable to establish benchmark datasets with ground-truth uncertainty. However, in real-world settings, ground truth uncertainty is usually unattainable. Thus related works have resorted to empirically obtained (and therefore not fully quantifiable and/or controllable) distribution shifts and label noise [20, 3], which has greatly advanced the field, albeit by construction still does not facilitate comprehensive insight into method behavior. Synthetic data generation offers a promising avenue towards improved insight by providing clearly defined data properties and annotations (see [17] for an example from the realm of Explainable AI). However, previous synthetic data generation methodologies proposed in the context of challenging image segmentation problems either excel in controllability but fall short in complexity [30, 38], or vice-versa aim at improved complexity and realism but at the cost of falling short in controllability [8, 40], the latter because learnt image generation, while able to offer some level of conditioning on sought image properties, neither provides full control nor full insight into the image generation process.

To address this gap, we introduce Arctique (ARtificial Colon Tissue Images for Quantitative Uncertainty Evaluation), a procedurally generated histopathological dataset designed to mirror the properties of images derived from H&E stained colonic tissue biopsies, as acquired routinely for safety-critical medical diagnoses in clinical practice [35]. Histopathological images offer a rich and challenging landscape for the application of advanced machine learning methodology, particularly in segmentation [2, 25]. The essential task of accurately delineating and classifying cellular structures is challenging even for trained professionals, due to many sources of uncertainty, e.g. overlapping structures, partial information from the underlying physical tissue-slicing process, and the inherent variability of biological tissues. The demanding nature of this task is reflected in the relative scarcity of fully annotated real-world data sets and high inter-annotator variability (see e.g [14]). Arctique offers the creation of realistic synthetic histopathological images at full controllability, allowing users to manipulate a range of easily interpretable parameters that effectively serve as "sliders" for image- as well as label uncertainties.

Arctique provides 50,000 pre-rendered 512x512-sized images for training and evaluation of segmentation tasks, shipped with exact masks (2D and 3D), metadata storing characteristics of cellular objects, and rendering parameters to re-generate scenes. Furthermore, Arctique provides two main avenues for the controlled study of uncertainty: (1) a blender-based generation framework, which allows to re-generate and manipulate scenes, and (2) a data loader for post-processing images and emulating noisy labels. To assess Arctique's degree of realism, we show that segmentation networks trained exclusively on Arctique can achieve promising zero-shot performance on real H&E images, proving its ability to learn meaningful attributes.

To showcase how Arctique can be used for insightful benchmarking of UQ methods, we assess foreground-background segmentation and semantic segmentation and measure the effect of uncertainty in the images and the labels separately. We benchmark the performance of four widely used UQ-methods (Maximum Softmax Response (MSR), Test Time Augmentation (TTA [37]), Monte-Carlo Dropout (MCD [11]) and Deep Ensembles (DE [22])). For each uncertainty scenario we measure model performance, predictive uncertainty, epistemic uncertainty and aleatoric uncertainty. Overall, we find that our manipulations generally increase predictive uncertainty in all four benchmarked UQ methods. In particular, we find that their aleatoric uncertainty components mostly track our devised label-level manipulations while their epistemic components mostly track our devised image-level manipulations. This serves as proof-of-concept that Arctique facilitates meaningful and comprehensive UQ benchmarking. Arctique was rendered and assessed on an internal resource of Nvidia A40 GPUs. Our work amounted to an estimated total of 150 GPU-hours.

**Dataset access** The current version of our dataset, as well as the complete version history, can be accessed via https://doi.org/10.5281/zenodo.11635056. We provide access to up to 50,000 training and 1,000 test images along with their corresponding instance and semantic masks, including 400 additional exemplary variations corresponding to 50 of the test images. We also provide the dataset used for the experimental results presented in this paper as well as the respective noisy variations. The complete codebase containing scripts for dataset generation, model training and uncertainty estimation is available on GitHub: https://github.com/Kainmueller-Lab/arctique

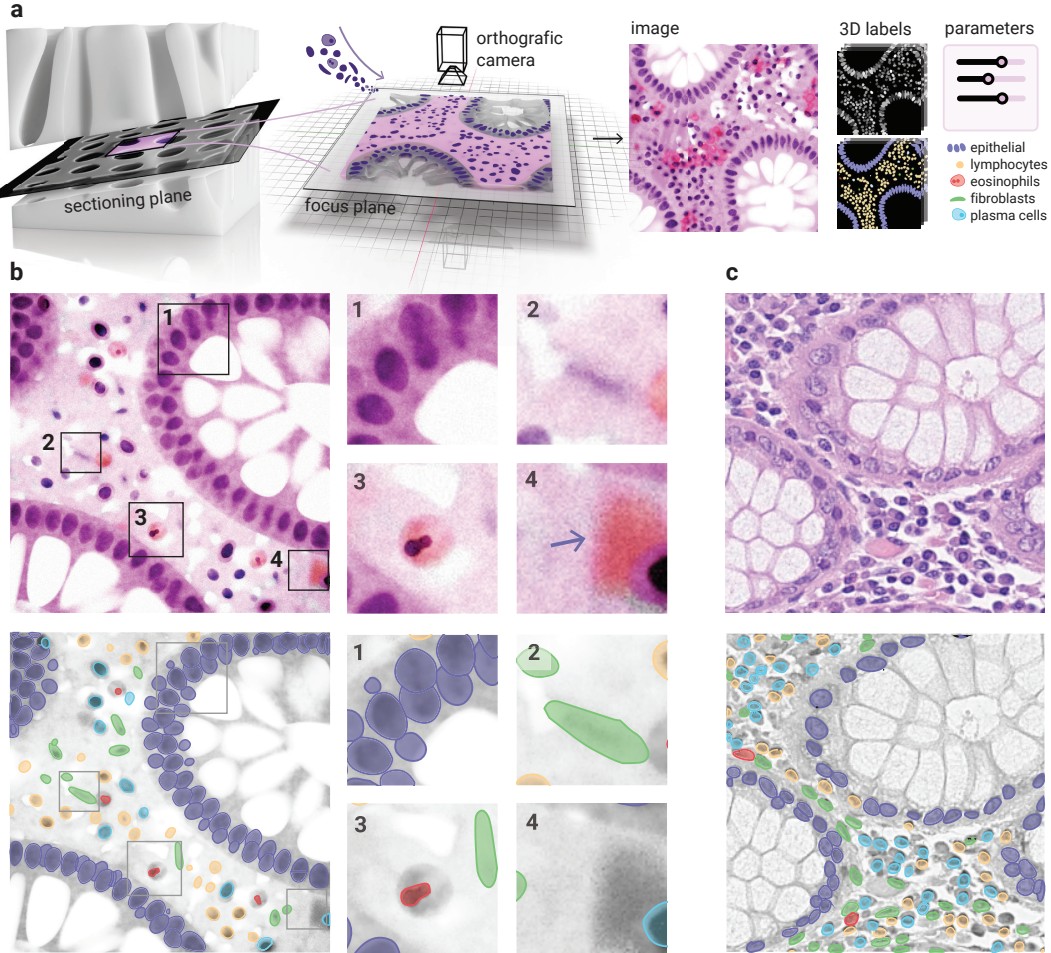

Figure 1: **Generation Process**: (a) To generate complex microscopic images, Arctique artificially replicates the H&E colon image creation protocol. From left to right: Initially, the colonic macrostructure (i.e., the outer epithelial layer) is constructed. This geometry is then artificially sliced, cell nuclei and other objects are placed, and the resulting scene is rendered along with its corresponding 3D stack of instance and semantic masks. (b) The result is a synthetic image (top) with corresponding semantic and instance mask (bottom) featuring numerous cell nuclei that (1) overlap, (2) lie outside the focal plane, (3) exhibit distinct characteristics, and (4) can be confused with perturbing elements. (c) A typical image of a natural H&E stained slice of colonic tissue (top) and the corresponding segmentation (bottom).The *epithelium* is characterized by its distinctive flower-like structures, known as *crypts*. The *stroma*, located between the crypts, forms the supportive connective tissue framework.

## 2   The Dataset

Histopathological Hematoxylin & Eosin (H&E) stained tissue slices captured under light microscopy pose a significant challenge for segmentation models. This is due to their inherent complexity, manifested by overlapping cells, varying staining intensities, limitations introduced by the physical tissue-slicing process, and general heterogeneity of biological tissues. The scarcity of exact annotated data further aggravates the problem, hindering the development of robust segmentation models. Our synthetic dataset Arctique addresses these challenges by mimicking the complexity of real H&E stained colon tissue images akin to those in the *Lizard* dataset [14]. To this end, we devised a Python-based image generation pipeline on top of the 3D ray-tracing software Blender. This approach, as opposed to the alternative of relying on generative models, ensures controllability (and reproducibility) while allowing for the creation of realistic scenes. Consequently, each of our

pre-rendered images and labels can be easily recreated and subjected to controlled modifications. We generate each H&E image along with its corresponding masks via the following procedural data generation pipeline, as shown in Figure 1a:

**1. Macro-structure:** We begin by generating a 3D model representing the characteristic topology of the colon tissue architecture. Specifically, we focus on the *epithelial crypts*, which do not only follow an intriguing pattern (see Figure 1a left) but are also of pathological relevance. For example, colon cancer typically originates at this outer layer of the colon. To model the outer tissue topology, we arrange rod-shaped crypts in a densely packed hexagonal pattern, as depicted in Figure 1a (for details see Appendix).

**2. Placing of cells:** Next, we populate our scene with five predominant cell types. Within the stroma, the connective tissue between the crypts, we randomly distribute plasma cells, lymphocytes, eosinophils, and fibroblasts. The cells of the epithelial crypts, i.e. epithelial cells and corresponding goblet cells (white bubbles, see Figure 1), are placed according to a 3D adaptation of the Voronoi cell generation algorithm (cf. [36, 23] and Appendix). Each cell model includes the nucleus and, when highlighted by staining, the surrounding cytoplasm. For instance, Figure 1b3 illustrates the peanut-shaped nucleus of an eosinophil within its reddish-stained cytoplasm. The individual cell types are characterized by controllable parameters such as typical diameter, elongation, and nucleus shape. A comprehensive description of the parameter sampling methodology is provided in the Appendix.

**3. Sectioning**: A significant source of complexity arises from the fact that histopathological images are 2D slices of an underlying 3D architecture. To replicate this, we digitally slice through our 3D macro structure and cells. This approach ensures that the visible features of cells vary depending on their location and orientation relative to the sectioning plane. For example, in Figure 1b2, we can faintly observe two fibroblasts: one cut along its major axis and another cut vertically.

**4. Staining:** In real-world histopathological images, the staining colors are derived from Hematoxylin & Eosin (H&E) staining. Hematoxylin binds to DNA in the nucleus, giving it a purple appearance, while eosin stains the surrounding tissue architecture in a reddish-purple hue (see Figure 1c). To replicate this, we model the staining of the cytoplasm, cell nuclei, and surrounding tissue using controllable parameters such as staining hue, staining intensity, and inherent staining intensity noise. This is achieved using Blender-specific shaders, as detailed in the Appendix.

**5. Rendering:** The final scene is rendered using ray tracing from a virtual camera positioned above the light source and tissue slice (see Appendix). By adjusting the camera's focal plane, we achieve a depth-blurring effect characteristic of histopathological light microscopy images. This workflow enables the generation of both 2D images and 3D image stacks. Moreover, the synthetic image generation provides precise 3D pixel-wise semantic- and instance masks and their corresponding 2D projection, serving as exact ground truth.

**Parameters:** Various parameters can be gradually adjusted to control the rendering process and allow for precise customization of the generated images: *Cell/Nuclei shapes*: Adjustments include cell diameter, elongation, bending, and shape noise for linear interpolation between cell types. *Cell distribution*: Parameters cover cell locations, occurrence ratios of cell types, and cell density in the stroma. *Tissue parameters*: Configurations include tissue thickness and degrees of tearing. *Staining parameters*: Settings include staining colors and intensities for cells, nuclei, and tissue. By consulting with a pathologist, we fine-tuned these parameters to align with the images from the Lizard dataset.

## 2.1 Assessment of Realism

Mimicking the generation protocol of real histopathological images, we were able to qualitatively incorporate many of their defining properties. Figure 1b demonstrates the fidelity of our dataset in capturing these characteristic nuances: 1b1) depicts a ring of densely packed and overlapping epithelial cell nuclei. Note that with Arctique, we enable a thorough investigation of such overlaps by offering precise 3D masks alongside their 2D projections, surpassing the limitations of standard 2D annotations typically used in real H&E slices; 1b2) illustrates the "blurring" effect of a fibroblast (characterized by its elongated shape) located outside the focus plane; and 1b3) showcases an artificial eosinophil cell, characterized by its distinctive peanut-shaped nucleus. Our dataset accurately models this characteristic feature, including the reddish staining hue of the surrounding cytoplasm, which contrasts with other cytoplasmic staining patterns. We also incorporated realistic sources of noise,

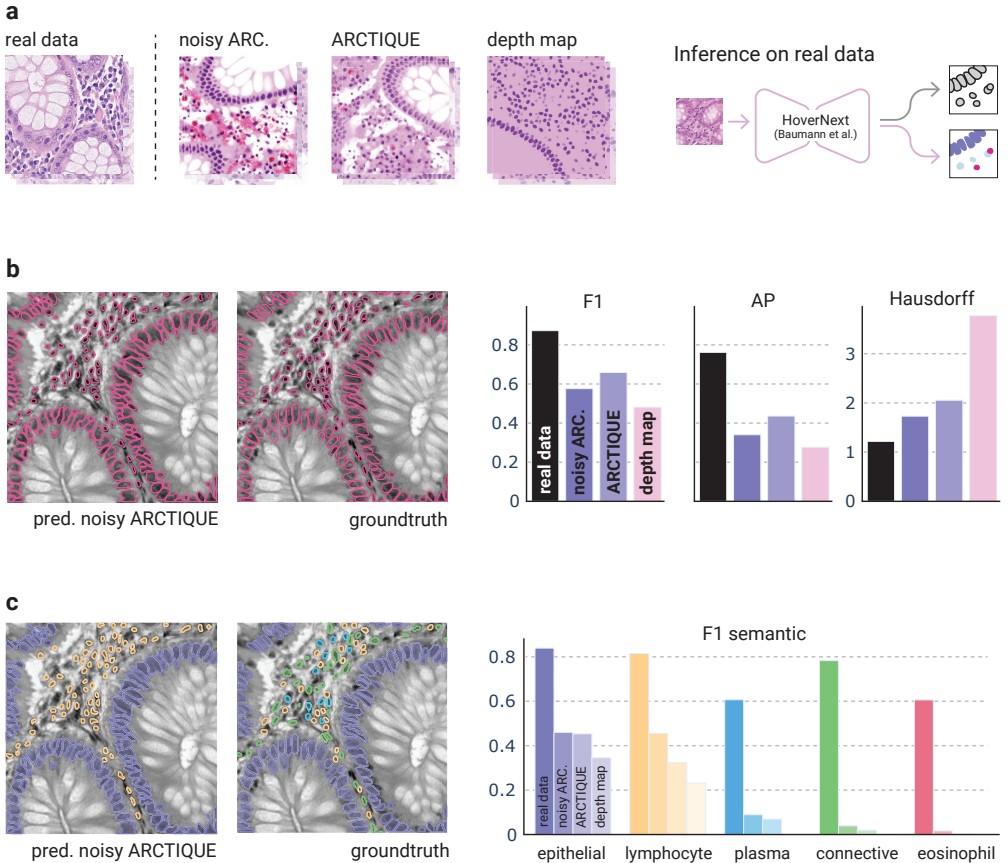

Figure 2: **Inference on the Lizard dataset using HoVer-NeXt (HN) models trained on Arctique:**
(a) Graphical illustration of the Arctique variants used for zero-shot learning, arranged on the left by
complexity level (from most to least complex and noisy). Each variant aims to enhance the model's
generalization across diverse structural and textural details. On the right, a schematic representation
depicts the post-processed raw class- and instance map outputs from the HN model during inference.
(b) and (c) show visual and quantitative results for instance- and semantic segmentation, respectively,
with bar plots comparing the baseline HN model trained on Lizard data (black) to the three HN
models trained on simulated datasets of varying complexity. All metrics and predictions are averaged
across 5 inference rounds, each with 16 Test-Time Augmentations. Note that the colors of the bars in
(c) correspond to the colors of celltypes in the example.

such as red blood cells, which exhibit a red staining hue similar to the cytoplasm of eosinophils (see
Figure 1b4).

**Zero-shot segmentation of real data:** To support these qualitative efforts, we quantitatively assess
the applicability of Arctique in a segmentation context by training a HoVer-NeXt (HN) model [4] on
Arctique. This panoptic segmentation architecture has been shown to yield state-of-the-art results
when trained on real H&E data. After training, we conduct zero-shot inference on real H&E data [14].
To validate Arctique's ability to infer semantically meaningful intermediate attributes, we compare
baseline results with a model trained a) on a "noisy" Arctique version containing randomly injected
anomalies, and b) on a simplified version consisting of depth maps only. As shown in Figure 2,
considering the instance segmentation task, both F1 score and Hausdorff distance metric (weighted for
true positives per class) support Arctique's realism and value. Qualitative tile inspections further reveal
that Arctique can detect previously discordant nuclei. Even regarding the semantic segmentation task,
the metrics per class indicate a positive correlation between predictions and observations for the most
abundant cell type, i.e. epithelial cells, without any fine-tuning. Overall (except for the default and
noisy version for F1 and AP score), the metrics exhibit a clear trend between increasing heterogeneity

in the synthetic data and better segmentation performance. In summary, these findings indicate that segmentation models trained on Arctique not only learn features pertinent to classical segmentation tasks, but also suggest that Arctique may serve as a promising surrogate training dataset. For further details on the training process, datasets description, and metrics comparison, see the Appendix.

## 3   Benchmarking uncertainty quantification methods

To showcase the capabilities of the Arctique dataset for benchmarking uncertainty quantification methods, we study the effect of image-level and label-level uncertainties on foreground-background segmentation (FG-BG-Seg) and semantic segmentation (Sem-Seg) [24]. To serve as a proof-of-concept, we evaluate the performance of established algorithms on our data, namely segmentation with a UNet backbone [32, 18], and uncertainty estimation with four popular methods, two ensemble-based, namely *Monte-Carlo Dropout* (MCD, [11]) and *Deep Ensembles* (DE, [22]), one heuristic, namely *Test Time Augmentation* (TTA, [37]), and, for comparative purposes, one deterministic model known as *Maximum Softmax Response* (MSR).[1] (See Appendix for implementation details).

In accordance with [20], we use the predictive entropy $\mathbb{H}[Y|x, \mathcal{D}]$, conditional on the training set $\mathcal{D} = \{x_i, y_i\}_{i=1}^n$, as the uncertainty measure of our predictive distribution $p(y|x, \mathcal{D})$, called *predictive uncertainty (PU)*. For all UQ models except MSR we estimate $p(y|x, \mathcal{D})$, by sampling from the models and averaging the softmax outputs over the samples. Following [7, 21], we can then perform an information-theoretic decomposition to disentangle, respectively, the epistemic and the aleatoric components. In this setting, the epistemic uncertainty is defined as the mutual information between the output $y$ and model parameters $\omega$:

$$\underbrace{\mathbb{H}[Y|x, \mathcal{D}]}_{\text{Predictive Unc. (PU)}} = \underbrace{\mathbb{I}[Y; \omega|x, \mathcal{D}]}_{\text{Epistemic Unc. (EU)}} + \underbrace{\mathrm{E}_{p(\omega|\mathcal{D})}[\mathbb{H}[Y|x, \omega]]}_{\text{Aleatoric Unc. (AU)}}. \qquad (1)$$

Eq. (1) shows that the aleatoric component correlates with the ambiguity inherent to the data and we should expect high values when there is a mismatch between image and label [21]. In particular, this implies that the aleatoric component is only meaningful for in-distribution data. The epistemic component, on the other hand, correlates with the model's lack of knowledge. It is sensitive to out-of-distribution (OOD) data and can be compensated for by the addition of new training data.

While the UQ measures yield pixel-wise results, we want to relate the uncertainty measures to our image- and label-level manipulations and are thus interested to aggregate pixel-level results to obtain uncertainty scores per image. It has been shown in [20] that the specific type of aggregation employed can hugely influence the behavior of uncertainty metrics. To account for this we tested the three aggregation strategies discussed in [20]: *image-level aggregation*, where uncertainty scores for all pixels are summed for each image and averaged over the dataset; *patch-level aggregation*, where uncertainty scores are aggregated within a sliding window and the maximum of the patch scores is taken as the image-level score; and *threshold-level aggregation*, which considers only uncertainty scores above an empirical quantile $\widehat{Q}_{u(\widehat{y})}(p)$ for a chosen uncertainty measure $u(\widehat{y})$, then calculates their mean. All results presented in the main text are generated using *threshold-level aggregation* and normalization based on the image size. In the Appendix, we provide results from alternative aggregation strategies for comparison.

In our experiments, we validate uncertainty measures using the variables defined in Eq. (1). Our approach differs from previous studies, such as [20, 29, 10], by focusing on how well information-theoretic definitions of aleatoric and epistemic uncertainty capture true uncertainty within our dataset, especially for complex tasks like semantic segmentation. While some studies, like [15, 26], examine epistemic uncertainty for segmentation, they are generally limited to in-distribution data. Additionally, we track prediction accuracy across varying noise levels, expecting accuracy to decrease as overall uncertainty increases.

---

[1]Where a deterministic model predicts a categorical distribution $p(y|x, \omega)$, we define $1 - \mathrm{MRS}$ as $1 - \max_c p(y = c|x, \omega)$, a metric employed as computationally cheap alternative to the predictive entropy [16] despite depending only on a single model realization (see also [27]).

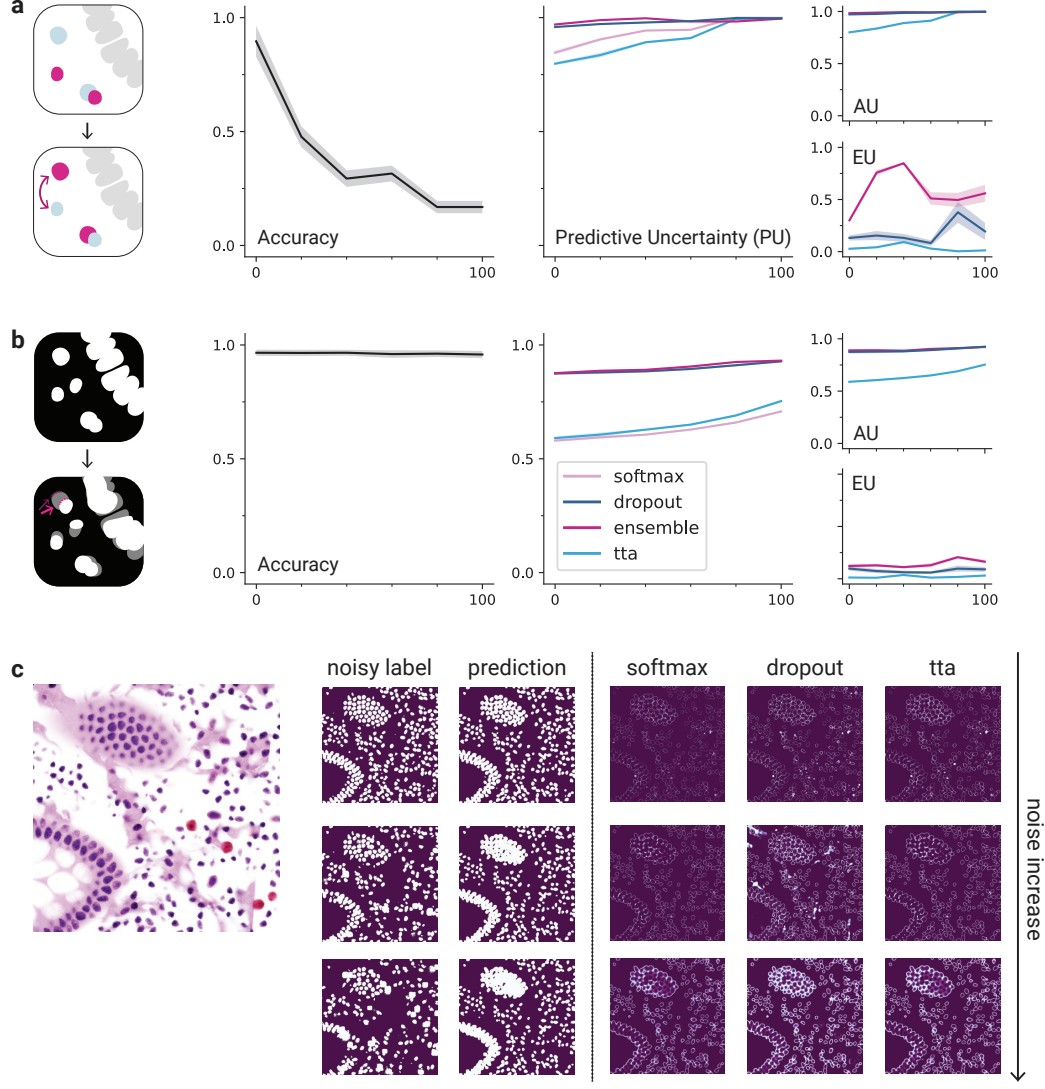

Figure 3: **Illustration of two types of label uncertainty and their effect on model performance and uncertainty measure.** (a) Effect of noisy class labels on Sem-Seg: illustrations on the left show an example of possible label confusion. The two large panels in the middle show model performance across noise levels (x-axis) as measured by accuracy and predictive uncertainty for all four UQ methods. The two smaller panels on the right show aleatoric and epistemic uncertainty for DE, TTA and MCD. (Note that MSR does not permit decomposition, therefore not shown.) (b) Effect of noisy label shapes on FG-BG-Seg: subpanels analogous to (a). (c) Qualitative example of the impact of noisy labels for FG-BG-Seg on prediction performance and how this is captured in the PU maps.

## 3.1 Label Noise

In the first step, we look at the effect of uncertain labels. In biomedical data, this is a common issue as complex and ambiguous images yield high disagreement even among expert annotators. In real-world images, we should expect some correlation between uncertainty in images and uncertainty in the labels. For example, cells with lower contrast staining might be harder to identify for human annotators leading to more missing labels. However, we believe, that there is a benefit to studying label-noise in isolation as it can give us valuable insights into model calibration and the sensitivity of UQ methods [6, 13, 19].

We devise two types of label-noise tailored to different segmentation tasks: *Sem-Seg:* Class labels are randomly switched. The noise level reflects the probability for each single-cell label to be switched to another class. *FG-BG-Seg:* Labels of single cells are manipulated by shifting, scaling, elastic transform or completely removed (missing label). The noise-level reflects the probability that any single cell is affected by any of these modifications. Both types of label noise are illustrated in the top row of Figure 3.

Figure 3 summarizes the results of uncertainty evaluation in the presence of label noise: for both segmentation tasks, we find that performance decreases with increasing label noise. This is to be expected as unreliable labels make it harder for the model to learn generalizable patterns. Predictive uncertainty (PU) increases as a function of label noise across both tasks. This confirms that what we would intuitively consider as "making the segmentation task harder" will actually decrease performance and increase uncertainty. Further, we find that across both segmentation tasks the majority of uncertainty stems from the aleatoric component. This is in keeping with the theoretical claim that aleatoric uncertainty mainly captures data-inherent uncertainty.

While aggregate measures offer a convenient way to assess how training with noisy labels impact a model's uncertainty, Arctique also provides exact labels, enabling a more detailed study of the label manipulation at pixel-level. Figure 3 shows in detail how models trained with and without label-noise learn about training images. Specifically, we observe that when some cell labels are consistently missing, the model still predicts the corresponding cells. Despite this, the uncertainty maps still highlight high uncertainty in regions with corrupted masks, suggesting that uncertainty quantification may address the common challenge of sparse annotations in biomedical images. Conversely, in densely packed regions, the model tends to interpolate across missing labels, as seen in the epithelial crypt in the bottom left of 3(c). Here, the uncertainty maps effectively capture the increased uncertainty in areas where cells are incorrectly merged. This duality underscores the value of evaluating UQ methods for their ability to handle both sparse and ambiguous label regions effectively.

## 3.2   Image Noise

The greatest advantage of having full control over the dataset creation is that it allows us to perform targeted manipulations to certain aspects of the image while leaving all others unchanged. We are thus able to create samples that gradually transform from near-OOD, where outlier and inlier classes are quite similar, to far-OOD, where an outlier is more distinct. This method of generating data is fundamentally different from other common strategies, such as applying augmentations like color shifts or blur [5, 32], where we achieve global manipulations which do not correlate with input features; or testing on images from a different domain [34] (e.g. data from a different organ) where the exact impact on image components is unknown and uncontrolled.

For the *Nuclei-Intensity* manipulation, we progressively reduce the staining of the cells' nuclei until they become less distinct from their surroundings, while preserving details in other regions and simulating real-world inconsistencies in staining. In contrast, an image-level reduction uniformly lowers contrast across the entire image, a simpler manipulation achievable with basic augmentation techniques and far from the intended use of Arctique. For the *Blood-Stain* manipulation, we gradually increase both the red stains and the number of blood cells, simulating realistic, extreme variations in blood cell abundance. This adjustment reflects a possible scenario in histology where red-stained artifacts may be mistaken for cell types like eosinophils, which naturally exhibit red staining.

Figure 4 shows examples for both types of image-level manipulations and their effects on model performance and uncertainty. Subfigure 4 (a) shows the impact of manipulating the nuclei-intensity on the FG-BG-Seg task. As might be expected, we observe a decrease in accuracy and an increase in the uncertainty measures as staining intensity decreases. While the aggregated effects may appear subtle, the error maps reveal a clearly visible decline in prediction performance: as staining weakens, the model starts to hallucinate cells in the tissue of the crypt-structures.

In Subfigure 4 (b) we illustrate the effect of manipulating the the blood-cells and -stains on the Sem-Seg task. Even with perfect masks, the model already tends to misclassify blood cells as eosinophils. As the abundance of blood cells increases, the number of false positives rises, leading to

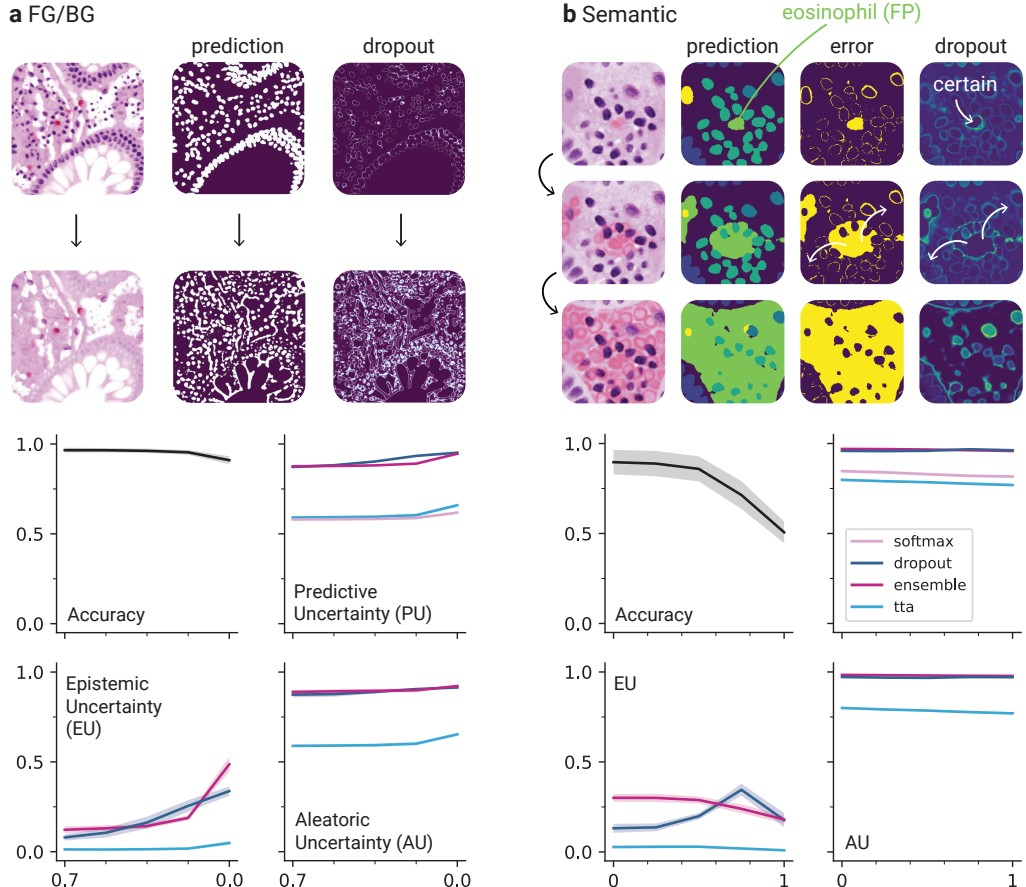

Figure 4: **Illustration of Image-level noise:** (a) Illustration of an image undergoing decreasing intensity of nuclei staining. The small image patches on the top illustrate qualitatively how FG-BG prediction performance and PU (for the example of MCD) are affected as staining is removed. The four panels on the bottom summarize for all four uncertainty methods how accuracy, PU, AU and EU react to the gradual change in staining. (b) illustrates the effect of the increasing prevalence of blood-cells. Similar as in (a) the small image patches on the top show the qualitative changes in semantic prediction performance and uncertainty. Here we additionally show the error maps next to the PU maps to highlight how blood cells are incorrectly identified as eosinophil cells, however the model remains confident in its prediction. The four panels on the bottom are arranged analogous to (a) and further illustrate the decrease in performance while uncertainty remains relatively unchanged.

a significant drop in accuracy, as shown in both the qualitative error maps and accuracy results in 4 (b). Consequently, this phenomenon leads to the miscalibration of uncertainty methods. In fact, the error maps reveal that regions affected by blood cells exhibit particularly low uncertainty values, indicating that high error rates do not correlate with high uncertainty. For DE we even observe a slight decrease in the uncertainty as the prevalence of blood cells increases, further emphasizing the weak correlation between error rates and uncertainty values.

We conclude that both experiments demonstrate Arctique's capabilities for in-depth analysis of uncertainty behavior. The two manipulations highlight common challenges encountered in real-world H&E images: without perfect labels, it becomes nearly impossible to ascertain whether high uncertainty values indicate subtle features in the data or stem from a miscalibrated uncertainty model. In the case of the *Nuclei-Intensity* alteration, the uncertainty method effectively identifies a genuine issue. In contrast, with the *Blood-Stain* manipulation, the uncertainty quantification (UQ) models demonstrate inadequacies in correctly calibrating the model.

# 4 Discussion

UQ carries the promise to increase the reliability of machine learning models so that these models can be more widely deployed even in safety-critical domains. To this end, we must be confident that the UQ methods we develop follow through on their claims. We believe that Arctique constitutes a valuable first step for a more thorough and interpretable evaluation of UQ metrics.

While the domain of histopathology may represent a specialized domain it serves as a valuable testbed due to its versatility and the presence of common uncertainty sources, such as missing or incorrect labels and overlapping instances. Moreover, this domain is particularly relevant for UQ, as it is critical for medical diagnosis and often suffers from incomplete and inaccurate annotations.

Our main goal in this publication is to introduce the Arctique dataset and illustrate its utility for evaluating UQ methods, yet it also opens numerous promising avenues for further research. One important follow-up would be to expand the range of studied UQ methods, particularly in Active Learning (AL), where uncertainty plays a central role in sampling strategies. Recent studies suggest that prioritizing high epistemic uncertainty can improve AL performance, while focusing on aleatoric uncertainty may be less effective ([6], [28]). Arctique's controlled uncertainty levels make it suitable for evaluating AL sampling, integrating uncertainty into optimization, and exploring domain adaptation strategies ([9], [39], [33]). In particular, Arctique allows to straightforwardly combine multiple sources of uncertainty at any level, thus constituting a unique testbed for methodology that seeks to disentangle AU and EU.

Our dataset can also be applied in the context of Explainable AI (XAI) evaluations, where transparent decision-making is crucial for trustworthiness. In contrast to simpler datasets like FunnyBirds [17], which focus on single-class tasks, Arctique offers a realistic multi-object environment. This complexity allows XAI methods to be benchmarked on relevant concepts that reflect the characteristics of real data, and to analyze interpretability and predictiveness across complex co-occurrence patterns, as for example cell nuclei and cytoplasm. Finally, future research could extend our framework with image- and label modifications, encompassing imaging modalities, tissue types, and staining variations. We encourage users to devise their own modifications suitable to their specific evaluation needs. A direct next step could be studying uncertainty for related tasks such as panoptic segmentation or 3D models.

To conclude, our work contributes Arctique, a complex, realistic yet fully controllable dataset of synthetic images, together with a broad range of "sliders" for targeted manipulation. As a proof-of-concept, we show that we can tailor label- and image manipulations such that they are selectively picked up by the aleatoric and epistemic components of established UQ methods, which suggests that Arctique is a valuable resource for UQ methods development and benchmarking, with clear potential for extensions into orthogonal methodological realms like XAI.

## Acknowledgments

We wish to thank Aasa Feragen, Kilian Zepf, Paul Jäger and Lorenz Rumberger for inspiring discussions. Funding: German Research Foundation (DFG) Research Training Group CompCancer (RTG2424), DFG Research Unit DeSBi (KI-FOR 5363, project no. 459422098), DFG Collaborative Research Center FONDA (SFB 1404, project no. 414984028), DFG Individual Research Grant UMDISTO (project no. 498181230), Synergy Unit of the Helmholtz Foundation Model Initiative, Helmholtz Einstein International Berlin Research School In Data Science (HEIBRiDS).

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
