# A  Appendix

## A.1  Tissue Modeling & Rendering

The Arctique dataset has been engineered to replicate the complexity of Histopathological Hematoxylin & Eosin (H&E) stained colon tissue images. This section describes the key steps involved in generating the synthetic dataset, encompassing macro-structure modeling, cell placement, sectioning, staining, and rendering.

**Macro-structure Modeling** We first generate a surface mesh object in Blender representing a large section of colonic tissue with intricate epithelial crypts to serve as the foundation for our dataset. Specifically, we mimic the rod-shaped crypts. These are arranged on a hexagonal grid, see figure 6a. To add more detail, the resulting mesh is then perturbed along the xy-plane using Blender's noise implementation. The crypts' surface mesh serves as basis to build the volumes where we will later place cell objects, namely the stroma (the space in-between the crypts) and ring-shaped volumes within the crypts for placing epithelial nuclei and goblets.

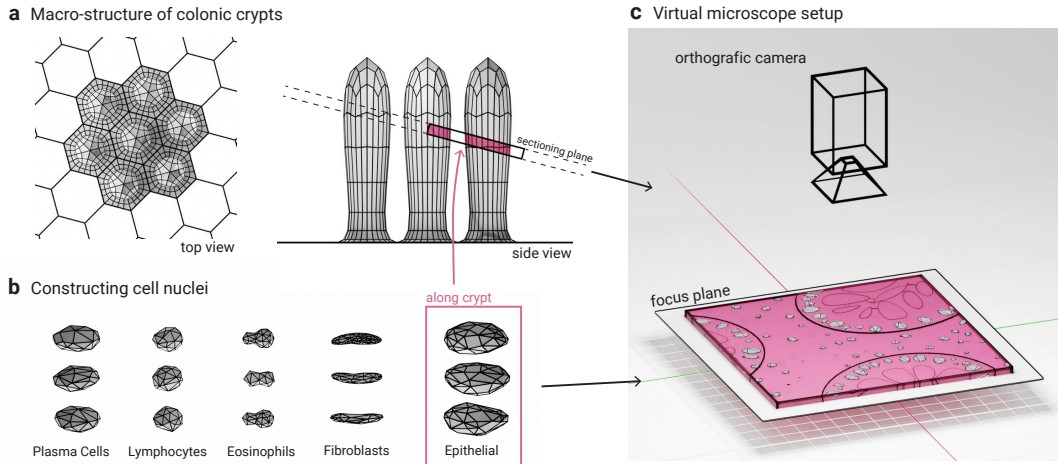

Figure 6: **Dataset Modeling** (a) The base surface of colonic tissue is modeled as multiple rod-shaped crypts organized in a hexagonal pattern. This model is then sectioned along a plane to produce a synthetic tissue sample. (b) Five common cell types and their nuclei are modeled and placed along the crypts. (c) After populating the sectioned tissue sample with cells and applying appropriate staining, a digital image is generated using a virtual microscope setup.

**Sectioning** To simulate the 2D nature of histopathological images, we generate smaller digital tissue sections akin to real-world tissue slices. We intersect the macro-structure volumes with a thin rectangular slice representing the real-world tissue dimensions (see figure 6a). This can result in sliced cells and nuclei, see figure 6c. The location and orientation of the sectioning plane is chosen randomly for each image resulting in a procedurally generated dataset with diverse appearances, see figure 6a.

**Cell Placement** From the sectioned macrostructure model we generate separate 3D volumes for the *stroma* and *epithelium*, which are then populated with procedurally generated cells, see figure 6b.

We model five common cell types, namely plasma cells, lymphocytes, eosinophils, fibroblasts and epithelial cells. Each cell consists of its nucleus and the surrounding cytoplasm and is characterized by controllable parameters such as its typical diameter, elongation, and nucleus shape. We first model each cell as an ellipsoid and then apply successive deformations using a random noise parameter until reaching a realistic cell shape. We distribute these cells uniformly throughout the stroma tissue.

The epithelium consists of an outer surface populated with epithelial cells and an inner surface populated with goblet cells which exhibit a characteristic ring- or flower-like structure when the tissue is sectioned. For this, we first sample points on the surface of the epithelium volume in order to achieve a slightly irregular hexagonal lattice structure. A cell type specific radius parameter hereby

controls the minimal lattice distance of these points. The points are then used as seeds to compute the vertices, egdes and faces of the resulting 3D Voronoi regions using the `scipy.spatial.Voronoi` library ([24], [16]). From these vertices, egdes and faces we generate a mesh for each polyhedral Voronoi region. The epithelial and goblet cells are then created by placing a deformed best-fitting ellipsoid into each Voronoi region.

**Staining** To replicate the characteristic staining colors of H&E images we use Blender's volumetric shaders. These mimic absorption and scattering in volumes with a set density. For individual objects such as cell nuclei, cell cytoplasm and the surrounding tissue we need to set the following parameters:

- **Staining Hue**: Specifies the color hue for the staining of the objects. For example, each cell type has a unique hue value to differentiate it visually in the images.

- **Staining Intensity**: Higher values result in more prominently stained cells, facilitating their identification and segmentation. Conversely, lighter staining can be used to introduce uncertainty at the image level, as it makes cell nuclei and cytoplasm less distinguishable from the surrounding tissue. Figure 7a shows an example of varying nuclei staining intensities.

- **Staining Noise**: This parameter helps to create more realistic synthetic images by adding slight variations to the staining across different cells. Such variability in the staining intensity can mimic real-world staining irregularities.

**Rendering** We simulate the lighting conditions of a light microscope by using an area light in Blender. This light source accurately mimics the illumination provided by a microscope's light. The final scene is then rendered by performing raytracing from a virtual camera placed above the light object and tissue slice, see figure 6c. By adjusting the camera's focal plane, we achieve a depth-blurring effect typical of histopathological light microscopy images. This workflow allows us to generate both 2D images and high-resolution 3D stacks. Additionally, the synthetic image generation provides precise pixel-wise semantic and instance masks, serving as exact ground truth for learning automatic segmentation tasks.

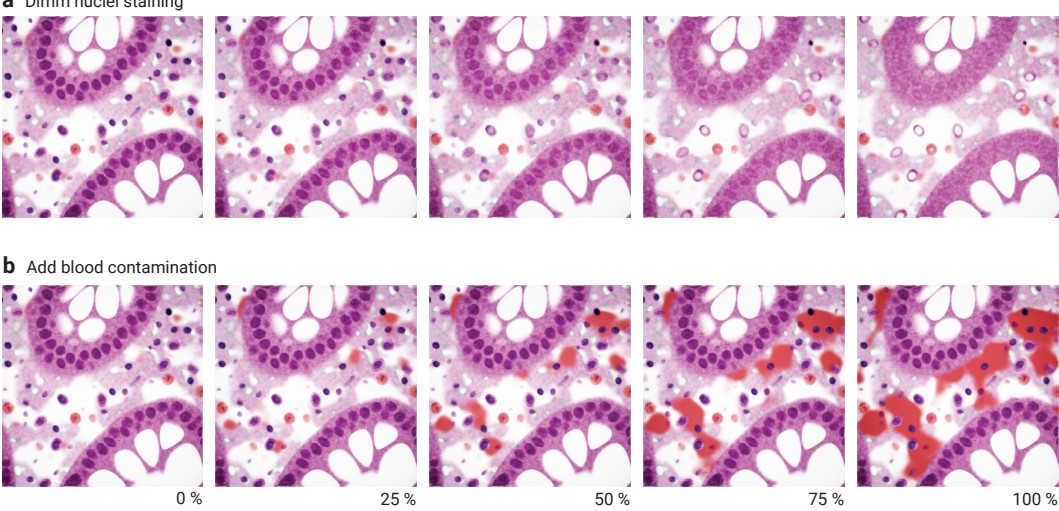

Figure 7: **Parameter-Sliders** allow to create different versions of the same scene. Contrary to widely used image augmentations such as e.g. blurring, here we only modify one component of the image while keeping the other parts of the image constant. (a) Successive reduction of nuclei staining which makes it harder to distinguish individuals cells from the background. (b) Gradually increased prevalence of blood-stain artifacts which can increase uncertainty when confused with cells.

**Controllable Parameters** The dataset generation process incorporates the following adjustable parameters to facilitate control over rendering outcomes. Many of these parameters allow to generate different variants of the same scene, as depicted in 7.

- **tissue thickness**: Controls the thickness of the digital tissue section. A thicker tissue allows for overlapping nuclei in the resulting image due to the higher depth.

- **tissue size**: This parameter sets the dimensions of the tissue area to be rendered.

- **tissue color**: Specifies the color of the tissue in RGBA format. This parameter defines the appearance of the tissue under simulated staining conditions.

- **tissue intensity**: Adjusts the color intensity of the tissue. Higher values result in more prominently stained tissue which makes nulcei harder to identify.

- **tissue ripping**: Simulates the ripping artifacts common to real-world scenarios. This parameter controls the occurence and strength of tissue ripping. Higher values introduce more irregularities in the tissue structure, mimicking real histological variations.

- **nuclei intensity**: Determines the staining intensity of cell nuclei. Higher values result in darker, more easily identifiable nuclei.

- **cell density**: Controls the density of cells within the stroma tissue.

- **cell type ratios**: Specifies the ratios of different cell types within the tissue. This parameter determines the relative abundance of lymphocites, plasma cells, eosinophils and fibroblasts.

- **cell shape mixing factor**: This parameter influences the blending of cell shapes across different cell types. Specifically, it enables linear interpolation of cell shapes between plasma cells and lymphocytes, allowing for controlled variation and uncertainty evaluation. This facilitates the creation of intermediate cell shapes, which can be useful for testing segmentation models under varying conditions.

## A.2 The Zero-Shot Segmentation

To support these qualitative efforts, we quantitatively assess the applicability of Arctique in a segmentation context by training a state-of-the-art model for panoptic segmentation, HoVer-NeXt (HN) [2], on Arctique. After training, we conduct zero-shot inference on real H&E colon tissue data. To validate Arctique's ability to infer semantically meaningful intermediate attributes, we compare the baseline results with those obtained from a model trained on Arctique variants, which feature either increased or decreased feature complexity, as detailed in Table 1.

| Symbol | Name | Description |
|--------|------|-------------|
| $\mathcal{L}$ | Lizard | H&E-based nuclei segmentation and classification dataset for large-scale colorectal cancer and normal colon tissues, providing segmentation masks for six cell types: neutrophils, epithelial cells, lymphocytes, plasma cells, eosinophils, and connective tissue cells [9]. It combines multiple datasets form several institutes and has $459179$ total annotated nuclei, but it is highly imbalanced, particularly with neutrophils and eosinophils. Additionally, $84\%$ of the dataset is background. Using a train-validation-test split of $80 - 10 - 10$, we create the benchmark datasets on which we train and evaluate the baseline model. |
| $\mathcal{A}$ | Arctique | Extensively described in Section 2 and A.1. For this experiment, a subset of $1450$ items was curated, split into training and validation sets with a $90 - 10$ ratio, with an additional test set of $50$ samples, all equipped with instance and semantic masks. |
| $\mathcal{A}_n$ | Noisy Arc. | Variant of $\mathcal{A}$. featuring more extreme parameter settings. Variations include increased blood cell density, modified nuclei intensity, altered epithelial cell size, overall hue shifts, and enhanced red tones in eosinophils. The sample count, image dimensions, and train-validation-test split remain consistent with $\mathcal{A}$. |
| $\mathcal{A}_{dm}$ | Depth Map | Variant of $\mathcal{A}$ with reduced complexity. Images are generated by mapping pixel values along the depth axis (from which the 3D image is sliced) to a dark color representing stained cell nuclei, while the background is assigned a color resembling surrounding tissue. The sample count, image dimensions, and train-validation-test split are identical to $\mathcal{A}$. |

Table 1: Description of the datasets used in the zero-shot learning experiment in Section 2.

**Model** HN builds upon the HoVer-Net model introduced in [10], simplifying the pipeline by replacing the binary nuclei segmentation map with a 3-class nuclei center-background prediction map (BCB-map) and merging two instance segmentation decoders into a single decoder. The architecture backbone is a U-Net [20] with an EfficientNet-V2 encoder [22]. HN further updates the model by incorporating a ConvNeXt-v2 encoder, known to deliver competitive results on various benchmarks [26]. In our experiments in Section 2, we utilize the ConvNeXt-v2 Tiny variant. Additionally, we replace the focal loss regularized by a sample-based class prior with a standard focal loss function [17].

**Training** For the experiments in Section 2, HN is trained for 200000 steps with a batch size of 12 using the AdamW optimizer with a weight decay of 0.0001. We employ a cosine-annealing learning rate schedule that ranges from $1e-4$ to $1e-8$. All encoders are trained with $50\%$ dropout, while the decoder does not use dropout.

Based on [2] and [23] we apply HED color augmentations, hue saturation and brightness variation, random noise and Gaussian blurring. We also include random rotation, flipping, mirroring , zoom, scale, shear, translate and elastic transform. Due to the specific sizes of the nuclei, masks and images of Arctique, in pre-processing phase the images were resized to $256 \times 256$ using nearest neighbor interpolation. This approach ensures that the features remain unchanged during training in relation to the size of the objects in the images. Additionally, the $\alpha$ parameter of the elastic transformation is magnified to enhance the realism of the cell borders.

The training loss combines the separate losses for instance and semantic segmentation, which are summed and weighted using a pre-set parameters ($\lambda = 0.02$). The individual loss components are:

- for the instance arm, the center point vector predictions are trained with MSELoss and the BCB-map with cross entropy loss;
- for the semantic arm, the Focal Loss (with parameter $\gamma = 2.0$) is employed.

Model selection is done via best validations metrics specific to the dataset instead of lowest validation loss.

**Inference** To reduce the risk of overfitting and enhance generalization, we apply Test-Time Augmentation (TTA) during both training and inference phases. Our TTA procedure includes HED color augmentation, mirroring, and $90°$ rotation to mitigate potential negative effects, such as those caused by excessive Gaussian blur.

HN enables tiles to be center-cropped and stitched together to create larger regions, facilitating parallel processing and metric computation. Based on individual class thresholds, we generate foreground areas and seed points in the BCB map, which are then processed using a watershed algorithm to extract nuclei instances. Small holes in instances are removed, and any false merges are addressed. The parameters previously validated for the Lizard dataset were reused to configure Arctique. Classes are assigned based on a majority vote, and instances are filtered according to class-specific size thresholds determined through hyperparameter search on the validation set.

**Evaluation** [5] argue that Panoptic Quality (PQ) should not be used for evaluating nuclei segmentation and classification because the small size of nuclei renders Intersection over Union (IoU) overly sensitive to coarse annotations. This sensitivity can lead to misleading evaluations, especially when the annotations lack precision. Given the intricate structures of nuclei, even slight misalignments in annotations can significantly impact IoU calculations, skewing the results and not accurately reflecting model performance. As a result, we do not report PQ for comparison, despite it being monitored during training and evaluation.

For binary detection we use F1 score and Matthews Correlation Coefficient (MCC). The detection method is based on the distance-based matching approach [21], an alternative to the widely used Intersection-over-Union (IoU). Then, we evaluate the detections using balanced accuracy and F1 Score. Detection metrics for Lizard and Arctique are evaluated on $248 \times 248$ center crops for consistency and to avoid having to detect nuclei with their center outside of the tile.

**Results** In the first experiment, we compare the baseline model, $\widehat{f}_{\mathcal{L}}$, trained and evaluated on the Lizard dataset, with a model $\widehat{f}_{\mathcal{A}}$ pre-trained on the Arctique dataset and inferred on Lizard. When subsequently compared to models trained on datasets with more complex features, $\widehat{f}_{\mathcal{A}_n}$, and less complex ones, $\widehat{f}_{\mathcal{A}_{dm}}$, the baseline model $\widehat{f}_{\mathcal{L}}$ achieves the highest overall F1 and AP score, followed by $\widehat{f}_{\mathcal{A}}$.

However, the Hausdorff distance and F1 semantic scores reveal a clear trend: increasing heterogeneity in the synthetic data correlates with improved segmentation performance, confirming the regularization effect of training on $\mathcal{A}_n$ [3]. For semantic segmentation, class-specific metrics for $\widehat{f}_{\mathcal{A}}$ and $\widehat{f}_{\mathcal{A}_n}$ show a positive correlation between predictions and ground truth for the most abundant cell type, without any fine-tuning.

The results, based on inference over 5 rounds with 16 TTA each, are presented in Figure 2.

### A.3 Segmentation Model for uncertainty quantification

**The Segmentation Backbone** For all the evaluated segmentation tasks, we use a standard U-Net [20] architecture with five convolutional blocks and ReLU activations. We add optional Dropout-layers after each convolutional block. We did not conduct exhaustive hyperparameter search, rather, we opted for default values as long as they produced reasonable results. Further exploration of more optimal training settings remains an open direction for future works.

The two segmentation tasks we address primarily differ in the number of output channels, while all perform pixel-wise predictions. For *FG-BG-Seg* the model has two output channels: one for the background (BG $= 0$) and one for the foreground (FG $= 1$). For *Sem-Seg* the model has five output channels: one for the background (BG $= 0$) and five for plasma cells, lymphocytes, eosinophils, fibroblasts and epithelial cells. [18]

All models were trained on Arctique, using the cross-entropy loss and the Adam optimizer, featuring a learning rate of $5\mathrm{e}{-}4$ and weight decay of $5\mathrm{e}{-}4$. The batch size is set to $24$ and the number of epochs to $200$. To ensure consistency in comparing UQ methods and performing zero-shot learning, each model incorporates dropout layers with a rate $p^{dropout} = 0.5$, and only flips and 90 degree rotations are used as augmentation during training. When studying the effects of noisy labels the training procedure was performed using segmentation masks with the desired degree of noise, while keeping all other training parameters identical.

### A.4 The Prediction Models

We test for different model predictions to derive corresponding measures of uncertainty, guided by the extensive UQ evaluation work outlined in [13]. Specifically, we compare four methods: *Monte-Carlo Dropout* [6], *Deep Ensembles* [15], *Test Time Augmentation* [25], and *Maximum Softmax Response* (MSR).

For all methods except Maximum Softmax Response, we generate $M$ probability maps $\mathrm{P}(\widehat{y}_{kj}^m = c)$, for each pixel coordinate $(k, j)$, by sampling $M$ realizations $\widehat{y}^m$ from the predictive distribution $p(\widehat{y}|x, \mathcal{D})$, which is inferred using the estimated set of weights $\widehat{\omega}$, and applying the Softmax function to the corresponding logits. In the following, we provide a more detailed outline of the sampling procedure for each of the UQ models.

**Monte-Carlo Dropout (MCD)** For the U-Net, dropout layers are activated during test time, and the model generates $M = 10$ predictions for each test image by passing it through the network 10 times. In each forward pass, the weight matrix is randomly masked, resulting in a distinct function being drawn for each prediction.

**Deep Ensembles (DE)** For the ensemble models, we train $M = 5$ U-Net backbones, each instantiated and trained according to the scheme described in A.3, but initialized with different random seeds. During test time, the image is passed through each of these models and their outputs are treated as samples from the predictive distribution. We select $M = 5$ as it offers a balanced trade-off between training cost and performance gain.

**Test Time Augmentation (TTA)** For the test time augmentations, we generate predictions by applying various combinations of vertical and horizontal flips, along with Gaussian blur, to each test input. Each augmented version, as well as the original unmodified input, is passed through the model to produce predictions. Prior to applying the Softmax function, inverse transformations are needed to align the predictions with the original object orientation [19]. Thereby, the process results in $M = 16$ forward passes (3 flipping possibilities, each with the same probability of occurrence and a small Gaussian noise).

### A.5 The Uncertainty Derivation

**Probabilistic approach** After generating the probability maps, we use the mean over the $M$ realizations as the prediction, $\bar{p}_{\widehat{y}_{kj}}(c) = M^{-1} \sum_m p_{\widehat{y}_{kj}}^m(c)^1$, and the (Shannon) entropy as the uncertainty measure, assuming the test-time inputs are independent. The pixel-wise predictive uncertainty is thus

---

[1]From this point forward, a more streamlined notation is used.

calculated as,

$$pu(\widehat{y}_{kj}) = \mathbb{H}[\widehat{y}_{kj}|x, \mathcal{D}] = -\sum_{c=1}^{C} \bar{p}_{\widehat{y}_{kj}}(c) \log \bar{p}_{\widehat{y}_{kj}}(c) \tag{2}$$

for the class labels $c = 1, \ldots, C$. Following [14], we compute the aleatoric component of the uncertainty based on the decomposition Eq.(1). This is achieved by calculating the entropy of each $m$-th pixel-wise prediction and then averaging across samples,

$$au(\widehat{y}_{kj}) = \mathrm{E}_{\widehat{\omega} \sim p(\omega|\mathcal{D})}[\mathbb{H}[\widehat{y}_{kj}|x, \widehat{\omega}]] = -M^{-1} \sum_{m=1}^{M} \sum_{c=1}^{C} p_{\widehat{y}_{kj}}^{m}(c) \log p_{\widehat{y}_{kj}}^{m}(c). \tag{3}$$

Finally, the epistemic component (or mutual information in Eq.(1)) is quantified through simple subtraction,

$$eu(\widehat{y}_{kj}) = pu(\widehat{y}_{kj}) - au(\widehat{y}_{kj}). \tag{4}$$

**Deterministic approach** MSR is an example of a deterministic model and therefore it is not possible to generate multiple samples from it. In this case, we define a computationally cheaper alternative to the predictive entropy [11] via Maximum Softmax Response,

$$pu^{msr}(\widehat{y}_{kj}) = 1 - msr(\widehat{y}_{kj}|x, \widehat{\omega}) = 1 - \max_{c} \mathrm{P}(\widehat{y}_{kj} = c|x, \widehat{\omega}) \tag{5}$$

where $\widehat{\omega}$ is a point estimate and not a random variable.

## A.6  A Note on Test Time Augmentation

The TTA model uniquely combines characteristics of both the deterministic and the probabilistic approaches. While it maintains the deterministic nature by not treating the parameter set $\widehat{\omega}$ as a random variable, it also embraces probabilistic elements by introducing diversity through stochastic transformations applied to the test inputs.

Under the same hypotheses as in [13], we consider a model that employs label-preserving transformations $T$, whose support is defined on the input space $\mathcal{T}$. This leads us to define the predictive distribution as $p(Y = y|x, \mathcal{D}) = \mathrm{E}_{t(x) \sim p(\mathcal{T})}[p(y|x, t(x), \mathcal{D})]$. In this context, the sampling procedure described in A.4 involves taking the expected value of the empirical predictive distribution $p(\widehat{y}|x, \mathcal{D})$ with respect to the distribution of transformations $p(t)$, rather than with respect to the distribution of model parameters $p(\widehat{\omega}|\mathcal{D})$.

Eq. (1) can thus be rewritten as,

$$\underbrace{\mathbb{H}[Y|x, \mathcal{D}]}_{\text{Predictive Unc. (PU)}} = \underbrace{\mathbb{I}[Y; t(x)|x, \mathcal{D}]}_{\text{Epistemic Unc. (EU)}} + \underbrace{\mathrm{E}_{p(\mathcal{T})}[\mathbb{H}[Y|x, t(x)]]}_{\text{Aleatoric Unc. (AU)}}. \tag{6}$$

Therefore, the expected pixel-wise entropy over the augmentations is supposed to give information about the amount of AU in the prediction for a new $x$,

$$au(\widehat{y}_{kj}) = \mathrm{E}_{t(x) \sim p(\mathcal{T})}[\mathbb{H}[\widehat{y}_{kj}|x, t(x)]] = -M^{-1} \sum_{m=1}^{M} \sum_{c=1}^{C} p_{\widehat{y}_{kj}}^{m}(c) \log p_{\widehat{y}_{kj}}^{m}(c), \tag{7}$$

and the mutual information between the augmentation variable $t(x)$ and the predicted label per pixel $\widehat{y}_{kj}$ can again be obtained using the formulation from Eq. (4). Given that our model remains invariant to transformations encountered during training [1], the mutual information between prediction and augmentation for a new test input would not be zero. However, if we were to augment the test image and include it in the training set for retraining, the mutual information would become zero. Hence, the term in Eq. (7) is reducible by adding new training points, thereby affirming the initial hypothesis that it represents epistemic uncertainty. This probabilistic approach tailored to TTA also facilitates the applications of the same inference functions delineated in A.5.

## A.7 The Aggregation Strategies

In order to compare uncertainty scores across different images, the pixelwise uncertainty maps have to be aggregated into a single numeric value. Motivated by the analysis in [13] we compare three different approaches for uncertainty aggregation: image-level, patch-level and threshold-level aggregation. Note that all results presented in Section 3 are based on threshold-level aggregation.

**Image level aggregation** Uncertainty scores for all pixels are summed per image and divided by the number of pixels. This simple approach is widely used in the literature, e.g [4, 8, 12]. However,[13] shows that for this approach the final uncertainty score correlates positively with the size of the segmented object.

**Patch level aggregation** employs a sliding window of size $10^d$ (where $d$ represents the image dimensionality) and averages pixel-wise uncertainties within the window. The final uncertainty score is determined by selecting the patch with the highest uncertainty. Based on the statistics of our dataset, we select a sliding window of $200$ pixels (roughly $14^2$).

**Threshold level aggregation** only takes the mean of the "most uncertain" pixels into account i.e. we only consider pixels with an uncertainty value above a given threshold. Following [13] the threshold is defined per image and takes into account the fraction of predicted foreground pixels.

First, the mean foreground ratio across all predicted segmentations in the validation set is so determined,

$$\alpha = \frac{\#\text{foreground prediction pixels}}{\#\text{pixels}} \tag{8}$$

The probability quantile is then calculated as $p = 1 - \alpha$, which is then used to compute the empirical quantile $\widehat{Q}_{u(\widehat{y}_{kj})}(p)$ for new predictions. Here $u$ can represent $pu(\widehat{y}_{kj})$, $au(\widehat{y}_{kj})$ or $eu(\widehat{y}_{kj})$. Empirical quantiles suffice for practical purposes. Finally we compute the average over all uncertainty scores that exceed $\widehat{Q}_{u(\widehat{y}_{kj})}(p)$.

For generating figure 3, 4 in Section 3, and figure **??**, 8, 9 in A.8, we compute as last step the average of the image-wise scores across the test dataset, assuming that the test images are independently and identically distributed (i.i.d.).

## A.8 Comparison of Alternative Aggregation Strategies

In this subsection, we present variations of figure 3 and 4 from Section 3, employing different aggregation strategies, namely *patch level* and *image level aggregation*.

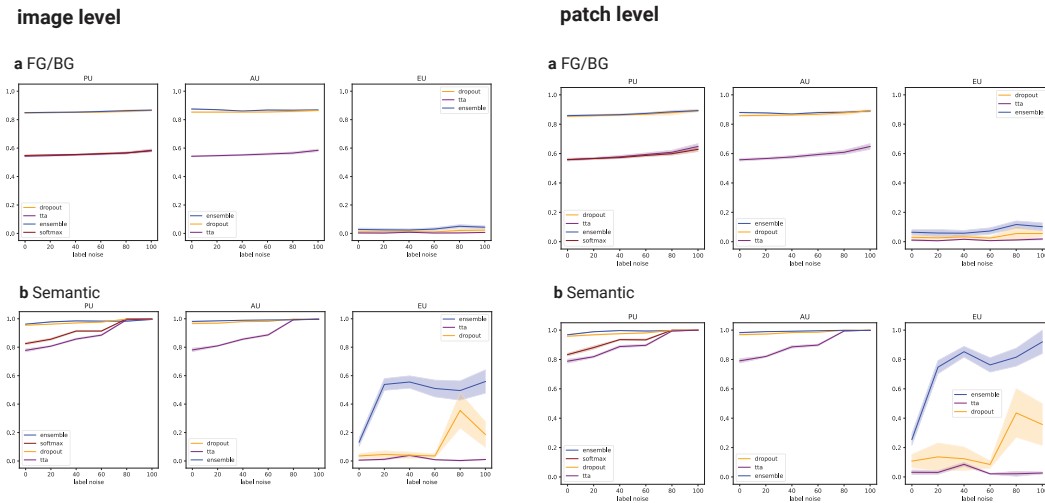

Figure 8: **Aggregation Strategies** Variant of figure 3 using image-level aggregation (left) and patch-level aggregation (right)

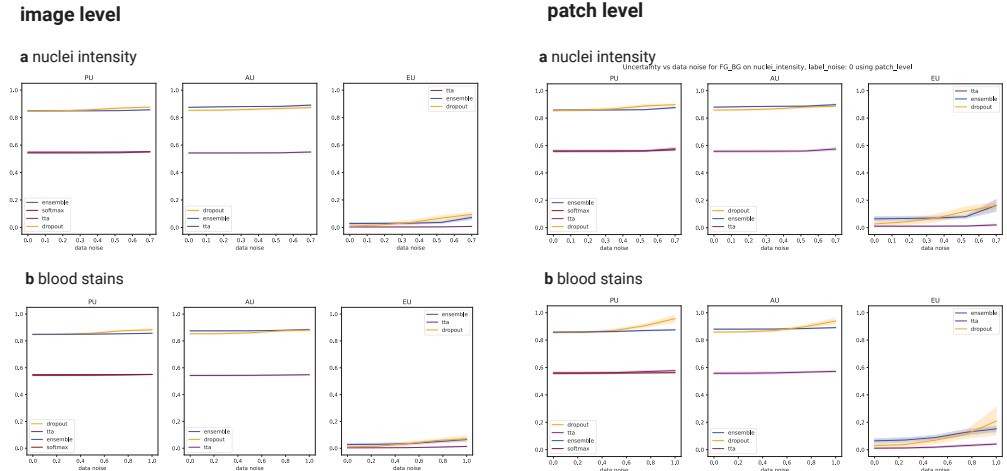

Figure 9: **Variant of figures 4(b) and 4(c) using image- and patch-level aggregation**. As in figure 4 we distinguish between decreasing intensity of nuclei staining (top) and increasing prevalence of red-spots (bottom). Sub-panels show the effect of the image-level manipulations for a FG-BG-Seg. model trained on exact labels.

### A.9 Applicability To Classification Tasks

While our primary experiments emphasize UQ for image segmentation, this section highlights the broader potential of our dataset. We focus on segmentation due to the significant challenges associated with obtaining accurate pixel-wise annotations. Nonetheless, the dense segmentation labels and procedural metadata in Arctique can also be leveraged to derive image-level labels, extending its applicability to classification tasks.

As a proof-of-concept, we generated binary labels from the segmentation masks to indicate the presence or absence of specific cell types. Utilizing Arctique's variational framework, we modified cell appearances and analyzed how these alterations influenced classification performance and uncertainty scores, see figure 10.

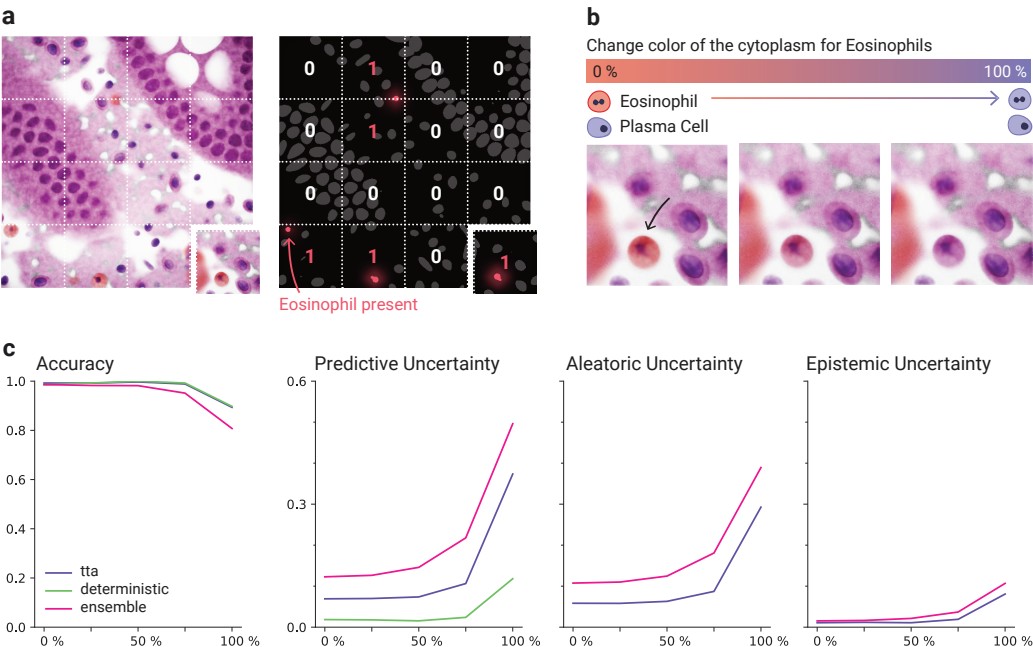

Figure 10: Arctique for Classification: From the existing segmentation masks we devise an exemplary classification task, namely detecting the presence/absence of eosinophils. a) Constructing image-level labels: We divide the image into smaller patches ensuring the desired cell type is not present on all patches. Then we use the mask to obtain binary class labels. b) We use the Arctique interface to vary the staining color of the eosinophilic cytoplasm from red to purple at test-time. c) As noise levels increase in this variation, accuracy declines, while both epistemic and aleatoric uncertainty increase.

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

## B   Dataset access

The current version of our dataset, as well as the complete version history, can be accessed via https://doi.org/10.5281/zenodo.11635056

In particular, the following link provides access to 50,000 training and 1,000 test images along with their corresponding instance and semantic masks, including 400 additional exemplary variations corresponding to 50 test images: https://zenodo.org/records/12704955

The dataset used for the results presented in this paper, including noisy variations, is provided here: https://zenodo.org/records/14016860

The complete code for reproducing the dataset, creating variations of the same dataset using parameter sliders and evaluating uncertainty quantification methods can be found here: https://github.com/Kainmueller-Lab/arctique

## C   Metadata access

Metadata for the dataset is provided by Zenodo, which offers export options in a variety of standard formats to facilitate easy integration and citation.

## D   Author Statement and Confirmation of Data License

The authors of this work bear all responsibility in case of any violation of rights, misuse of data, or other related issues.

Our final contribution encompasses three key components: the complete synthetic dataset, the codebase responsible for generating this dataset, and the codebase utilized for evaluation purposes. In adherence to Blender's GNU General Public License (GPL), we specify the following licensing arrangements:

**Dataset & Evaluation Codebase**: MIT License
The dataset itself and the codebase utilized for evaluation purposes will be licensed under the MIT License. This license allows for maximum flexibility and ease of use, enabling others to freely utilize, modify, and distribute the dataset and associated evaluation codebase without significant restrictions.

**Codebase for Image Generation**: GNU General Public License v3.0 (GPLv3)
The codebase responsible for generating the synthetic images interacts directly with Blender's API, thus forming a derivative work of Blender. Therefore, in accordance with Blender's GPL license, this codebase must also be distributed under the terms of the GNU General Public License v3.0 (GPLv3). This ensures that modifications and distributions of the image generation codebase are also subject to the principles of copyleft, ensuring the continued openness and availability of the codebase.

# E Dataset Documentation

In this section we answer the Datasheet for Datasets questionnaire [7] to document the Arctique dataset. It contains information about motivation, composition, collection, preprocessing, usage, licensing as well as hosting and maintenance plan.

## E.1 Motivation

**For what purpose was the dataset created? Was there a specific task in mind? Was there a specific gap that needed to be filled? Please provide a description.**
The dataset was created to simulate histopathological images for the purpose of developing and evaluating segmentation models. The specific task in mind was to create a dataset that reflects the complexity of Hematoxylin & Eosin (H&E) stained colon tissue in light microscopy, addressing the need for a reliable and reproducible dataset that can be used for training and testing machine learning models in this domain.

**Who created the dataset (e.g., which team, research group) and on behalf of which entity (e.g., company, institution, organization)?**
This dataset was created in a collaboration of the Kainmueller Lab from the Max-Delbrueck-Center for Molecular Medicine in the Helmholtz Association (MDC) and Helmholtz Imaging in Berlin, as well as the Institute of Pathology from the Charité, Universitätsmedizin Berlin, Corporate Member of Freie Universität Berlin and Humboldt-Universität zu Berlin.

**Who funded the creation of the dataset? If there is an associated grant, please provide the name of the grantor and the grant name and number.**
Funding: German Research Foundation (DFG) Research Training Group CompCancer (RTG2424), DFG Research Unit DeSBi (KI-FOR 5363, project no. 459422098), DFG Collaborative Research Center FONDA (SFB 1404, project no. 414984028), DFG Individual Research Grant UMDISTO (project no. 498181230), Synergy Unit of the Helmholtz Foundation Model Initiative, Helmholtz Einstein International Berlin Research School In Data Science (HEIBRiDS).

## E.2 Composition

**What do the instances that comprise the dataset represent (e.g., documents, photos, people, countries)? Are there multiple types of instances (e.g., movies, users, and ratings; people and interactions between them; nodes and edges)? Please provide a description.**
The instances in the dataset are digitally generated images akin to histopathological images that represent various cell types within colonic tissue. There are multiple types of instances, including images of tissue sections and corresponding pixel-wise semantic and instance masks.

**How many instances are there in total (of each type, if appropriate)?**
The full dataset includes 50,000 training images and 1,000 test images, each with corresponding instance and semantic masks, along with 400 additional variations based on 50 test images. Additionally, we provide the specific dataset used in our experiments, comprising 1,500 synthetic images without noise, 1,500 with added noise, and 1,500 depth mask images, along with a range of noisy variations.

**Does the dataset contain all possible instances or is it a sample (not necessarily random) of instances from a larger set? If the dataset is a sample, then what is the larger set? Is the sample representative of the larger set (e.g., geographic coverage)? If so, please describe how this representativeness was validated/verified. If it is not representative of the larger set, please describe why not (e.g., to cover a more diverse range of instances, because instances were withheld or unavailable).**
The dataset is a synthetic representation and not a sample from a larger set. It is designed to simulate the diversity and complexity of real-world histopathological images.

**What data does each instance consist of? "Raw" data (e.g., unprocessed text or images) or features? In either case, please provide a description.**
Each instance consists of raw image data of tissue sections, along with ground-truth pixel-wise semantic and instance masks.

**Is there a label or target associated with each instance? If so, please provide a description.**
Yes, each instance has corresponding ground-truth pixel-wise semantic and instance masks that serve as labels.

**Is any information missing from individual instances? If so, please provide a description, explaining why this information is missing (e.g., because it was unavailable). This does not include intentionally removed information, but might include, e.g., redacted text.**
No information is missing from the individual instances.

**Are relationships between individual instances made explicit (e.g., users' movie ratings, social network links)? If so, please describe how these relationships are made explicit.**
Relationships between individual instances are not applicable in this dataset.

**Are there recommended data splits (e.g., training, development/validation, testing)? If so, please provide a description of these splits, explaining the rationale behind them.**
The dataset can be split into training, validation, and testing sets as needed. The specific splits are left to the user's discretion based on their experimental setup.

**Are there any errors, sources of noise, or redundancies in the dataset? If so, please provide a description.**
The dataset is synthetically generated to minimize errors and noise. However, simulated noise and artifacts are intentionally included to reflect real-world conditions.

**Is the dataset self-contained, or does it link to or otherwise rely on external resources (e.g., websites, tweets, other datasets)? If it links to or relies on external resources, a) are there guarantees that they will exist, and remain constant, over time; b) are there official archival versions of the complete dataset (i.e., including the external resources as they existed at the time the dataset was created); c) are there any restrictions (e.g., licenses, fees) associated with any of the external resources that might apply to a dataset consumer? Please provide descriptions of all external resources and any restrictions associated with them, as well as links or other access points, as appropriate.**
The dataset is self-contained and does not rely on external resources.

**Does the dataset contain data that might be considered confidential (e.g., data that is protected by legal privilege or by doctor–patient confidentiality, data that includes the content of individuals' non-public communications)? If so, please provide a description.**
No, the dataset does not contain confidential data.

**Does the dataset contain data that, if viewed directly, might be offensive, insulting, threatening, or might otherwise cause anxiety? If so, please describe why.**
No, the dataset does not contain such data.

**Does the dataset identify any subpopulations (e.g., by age, gender)? If so, please describe how these subpopulations are identified and provide a description of their respective distributions within the dataset.**
No, the dataset does not identify subpopulations.

**Is it possible to identify individuals (i.e., one or more natural persons), either directly or indirectly (i.e., in combination with other data) from the dataset? If so, please describe how.**
No, it is not possible to identify individuals from the dataset.

**Does the dataset contain data that might be considered sensitive in any way (e.g., data that reveals race or ethnic origins, sexual orientations, religious beliefs, political opinions or union memberships, or locations; financial or health data; biometric or genetic data; forms of government identification, such as social security numbers; criminal history)? If so, please provide a description.**
No, the dataset does not contain sensitive data.

### E.3 Collection Process

**How was the data associated with each instance acquired? Was the data directly observable (e.g., raw text, movie ratings), reported by subjects (e.g., survey responses), or indirectly inferred/derived from other data (e.g., part-of-speech tags, model-based guesses for age or language)? If the data was reported by subjects or indirectly inferred/derived from other data,**

**was the data validated/verified? If so, please describe how.**
The data for each instance was directly observable, consisting of synthetically generated images of digitally modeled colonic tissue using the rendering software Blender. Validation was performed by comparing generated images with real histopathological images to ensure similarity.

**What mechanisms or procedures were used to collect the data (e.g., hardware apparatuses or sensors, manual human curation, software programs, software APIs)? How were these mechanisms or procedures validated?**
The image generation was carried out using the 3D rendering software Blender and Python scripts. The procedure was validated by comparing the generated images with real histopathological images to ensure visual and structural similarity.

**If the dataset is a sample from a larger set, what was the sampling strategy (e.g., deterministic, probabilistic with specific sampling probabilities)?**
The dataset is not a sample from a larger set; it consists entirely of synthetically generated images. Moreover, additional images can be produced using scripts with adjustable parameters.

**Who was involved in the data collection process (e.g., students, crowdworkers, contractors) and how were they compensated (e.g., how much were crowdworkers paid)?**
The data collection process was carried out by a team of researchers. Compensation details are not applicable as this was part of their research and development activities.

**Over what timeframe was the data collected? Does this timeframe match the creation timeframe of the data associated with the instances (e.g., recent crawl of old news articles)? If not, please describe the timeframe in which the data associated with the instances was created.**
The timeframe for generating the dataset depends on the available computing resources. Specifically, the image generation process alone spanned about one week, while the entire script generation process extended over approximately six months.

**Were any ethical review processes conducted (e.g., by an institutional review board)? If so, please provide a description of these review processes, including the outcomes, as well as a link or other access point to any supporting documentation.**
No ethical review processes were conducted as the dataset does not involve human subjects or personal data.

**Did you collect the data from the individuals in question directly, or obtain it via third parties or other sources (e.g., websites)?**
The data was not collected from individuals; it was synthetically generated.

**Were the individuals in question notified about the data collection? If so, please describe (or show with screenshots or other information) how notice was provided, and provide a link or other access point to, or otherwise reproduce, the exact language of the notification itself.**
Not applicable, as the dataset does not involve human subjects.

**Did the individuals in question consent to the collection and use of their data? If so, please describe (or show with screenshots or other information) how consent was requested and provided, and provide a link or other access point to, or otherwise reproduce, the exact language to which the individuals consented.**
Not applicable, as the dataset does not involve human subjects.

**If consent was obtained, were the consenting individuals provided with a mechanism to revoke their consent in the future or for certain uses? If so, please provide a description, as well as a link or other access point to the mechanism (if appropriate).**
Not applicable, as the dataset does not involve human subjects.

**Has an analysis of the potential impact of the dataset and its use on data subjects (e.g., a data protection impact analysis) been conducted? If so, please provide a description of this analysis, including the outcomes, as well as a link or other access point to any supporting documentation.**
Not applicable, as the dataset does not involve human subjects.

### E.4 Preprocessing/cleaning/labeling

**Was any preprocessing/cleaning/labeling of the data done (e.g., discretization or bucketing, tokenization, part-of-speech tagging, SIFT feature extraction, removal of instances, processing**

**of missing values)? If so, please provide a description. If not, you may skip the remaining questions in this section.**
Yes, preprocessing and labeling were done. After generating the synthetic images, exact pixel-wise semantic and instance masks were created to label each cell type accurately. Additionally, the images were reviewed and minor cleaning steps were performed to remove any artifacts from the rendering process.

**Was the "raw" data saved in addition to the preprocessed/cleaned/labeled data (e.g., to support unanticipated future uses)? If so, please provide a link or other access point to the "raw" data.**
There is no "raw" data as the images and labels were generated algorithmically.

**Is the software that was used to preprocess/clean/label the data available? If so, please provide a link or other access point.** No such software was employed.

## E.5   Uses

**Has the dataset been used for any tasks already? If so, please provide a description.**
Yes, the dataset has been used for training and evaluating segmentation models for histopathological images. The generated images and masks help in benchmarking and improving the performance of these models.

**Is there a repository that links to any or all papers or systems that use the dataset? If so, please provide a link or other access point.**
No, there is no such repository.

**What (other) tasks could the dataset be used for?**
The dataset could be used for a variety of tasks including but not limited to:
- Training and evaluating segmentation and classification models (both 2D and 3D)
- Evaluating uncertainty quantification methods
- Evaluating Explainable AI (XAI) methods
- Evaluating sampling strategies for Active Learning (AL)
- Research on domain adaptation in histopathological imaging

**Is there anything about the composition of the dataset or the way it was collected and prepro-cessed/cleaned/labeled that might impact future uses? For example, is there anything that a dataset consumer might need to know to avoid uses that could result in unfair treatment of individuals or groups (e.g., stereotyping, quality of service issues) or other risks or harms (e.g., legal risks, financial harms)? If so, please provide a description. Is there anything a dataset consumer could do to mitigate these risks or harms?**
The dataset is synthetically generated, ensuring no real patient data is involved, thus avoiding privacy concerns. However, users should be aware that while the dataset is designed to closely mimic real histopathological images, it may not capture all the nuances of actual tissue samples. Ensuring models trained on this data are validated on real-world data is crucial to mitigate potential biases.

**Are there tasks for which the dataset should not be used? If so, please provide a description.**
The dataset should not be used for any applications requiring actual patient data or clinical decision-making without further validation. It is also not suitable for tasks that require genetic or molecular level analysis as the dataset does not contain such information.

**Any other comments?**
The synthetic nature of the dataset allows for extensive control and reproducibility, making it a valuable resource for research and development in medical imaging. However, users should complement their studies with real-world data to ensure the applicability and robustness of their models.

## E.6   Distribution

**Will the dataset be distributed to third parties outside of the entity (e.g., company, institution, organization) on behalf of which the dataset was created?**
Yes, the dataset will be distributed to third parties outside of the entity to encourage further research and development.

**How will the dataset be distributed (e.g., tarball on website, API, GitHub)? Does the dataset have a digital object identifier (DOI)?**
The dataset including version history is publicly available via Zenodo. The complete code for reproducing the dataset, creating variations of the same dataset using parameter sliders and evaluating uncertainty quantification methods is available on GitHub. See Section B for URLs.

**When will the dataset be distributed?**
The dataset will be published and distributed upon acceptance of the paper.

**Will the dataset be distributed under a copyright or other intellectual property (IP) license, and/or under applicable terms of use (ToU)?**
Yes, the dataset will be distributed under the MIT license upon publication. There are no fees associated with this license.

**Have any third parties imposed IP-based or other restrictions on the data associated with the instances?**
No third parties have imposed IP-based or other restrictions on the data associated with the instances.

**Do any export controls or other regulatory restrictions apply to the dataset or to individual instances?**
No export controls or other regulatory restrictions apply to the dataset or to individual instances.

## E.7 Maintenance

**Who will be supporting/hosting/maintaining the dataset?**
The dataset will be supported, hosted, and maintained by the research team that created it.

**How can the owner/curator/manager of the dataset be contacted (e.g., email address)?**
Please reach out to the corresponding authors.

**Is there an erratum? If so, please provide a link or other access point.**
There is no erratum currently available. Any future errata will be posted on the dataset's GitHub repository which will be published upon acceptance.

**Will the dataset be updated (e.g., to correct labeling errors, add new instances, delete instances)? If so, please describe how often, by whom, and how updates will be communicated to dataset consumers (e.g., mailing list, GitHub)?**
The dataset will be maintained and kept up-to-date using version control in Zenodo. All updates and new versions will be communicated through Zenodo and the GitHub repository. See Section B for URLs.

**If the dataset relates to people, are there applicable limits on the retention of the data associated with the instances (e.g., were the individuals in question told that their data would be retained for a fixed period of time and then deleted)? If so, please describe these limits and explain how they will be enforced.**
This dataset does not relate to people, so there are no applicable limits on the retention of the data.

**Will older versions of the dataset continue to be supported/hosted/maintained? If so, please describe how. If not, please describe how its obsolescence will be communicated to dataset consumers.**
Older versions of the dataset will not be actively maintained, but they will remain accessible through Zenodo. Obsolescence of older versions will be communicated to dataset consumers via Zenodo.

**If others want to extend/augment/build on/contribute to the dataset, is there a mechanism for them to do so? If so, please provide a description. Will these contributions be validated/verified? If so, please describe how. If not, why not? Is there a process for communicating/distributing these contributions to dataset consumers? If so, please provide a description.**
We encourage others to reach out to our research group to contribute to the dataset. All contributions will be validated and verified by our team before integration. Updates and contributions will be communicated to dataset users through the repository.

# F   Dataset Format and Reading Explanation

## F.1   Dataset Structure

The Arctique dataset includes 50,000 training images and 1,000 test images, each with corresponding instance and semantic masks, along with 400 additional variations based on 50 test images. It is split into training and test sets, each containing directories for images and masks. There is an additional directory.

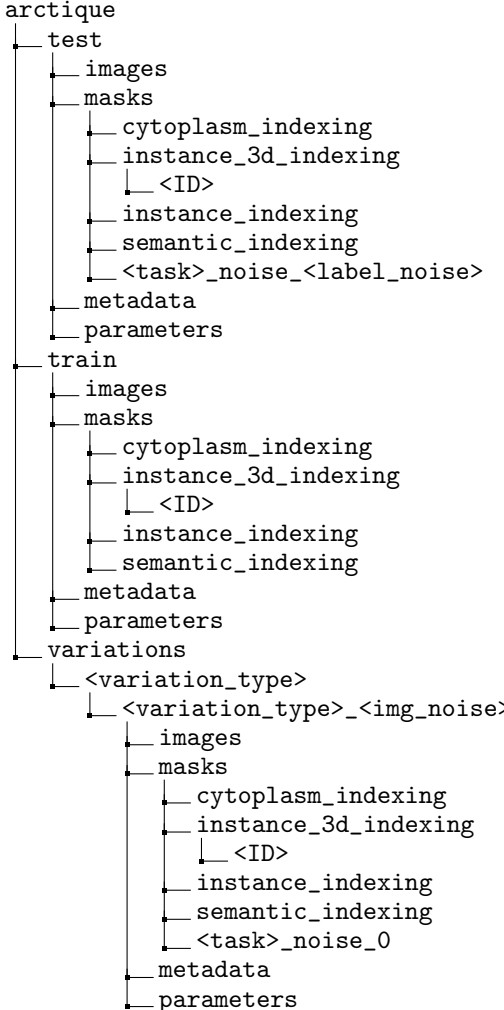

```
arctique
├── test
│   ├── images
│   ├── masks
│   │   ├── cytoplasm_indexing
│   │   ├── instance_3d_indexing
│   │   │   └── <ID>
│   │   ├── instance_indexing
│   │   ├── semantic_indexing
│   │   └── <task>_noise_<label_noise>
│   ├── metadata
│   └── parameters
├── train
│   ├── images
│   ├── masks
│   │   ├── cytoplasm_indexing
│   │   ├── instance_3d_indexing
│   │   │   └── <ID>
│   │   ├── instance_indexing
│   │   └── semantic_indexing
│   ├── metadata
│   └── parameters
└── variations
    └── <variation_type>
        └── <variation_type>_<img_noise>
            ├── images
            ├── masks
            │   ├── cytoplasm_indexing
            │   ├── instance_3d_indexing
            │   │   └── <ID>
            │   ├── instance_indexing
            │   ├── semantic_indexing
            │   └── <task>_noise_0
            ├── metadata
            └── parameters
```

## F.2   Dataset description

### Images

The `images` directory contains all synthetically generated images stored as PNG files. Each image has a resolution of 512x512 pixels with RGB channels and is named "img_<ID>", where <ID> is a unique integer identifier for each image.

### Masks

The `masks` directory includes subdirectories containing various masks related to the images.

- `cytoplasm`: Contains 2D semantic masks for the cell cytoplasm. Each mask corresponds to an image named "<ID>.tif", where "<ID>" is the identifier for that image. The mask file is named using the same identifier.

- `instance_3d`: Contains a directory for each image, named "<ID>". Inside each directory, there is a 3D stack numpy file representing the instance IDs in a 3D volumetric array. Additionally, it includes a sequence of 2D instance segmentation masks, named "slice_<ID>_<slice_count>.png", each representing equidistant slices through the 3D volume along the depth axis.

- `instance`: Contains 2D instance masks for the cell nuclei. Each mask corresponds to an image named "<ID>.tif", and the mask file is named with the same identifier.

- `semantic`: Contains 2D semantic masks for the cell nuclei. Similar to the instance masks, each mask corresponds to an image named "<ID>.tif", with the mask file named using the same identifier.

- `<task>_noise_<label_noise>`: Contains 2D masks for the cell nuclei, where individual masks have been deformed and/or removed according to a specified probability, indicated by "<label_noise>". These modifications align with the specific requirements of the task labeled as "<task>". The `<task>_noise_0` directory consistently contains the original, unmodified masks. Each mask corresponds to an image named "<ID>.png", with the mask file named using the same identifier.

Note that all semantic masks appear as black images when viewed with a standard image viewer. This is because the cell type IDs, ranging from 1 to 6, are used as greyscale values, which appear dark in the images.

**Metadata**

The `metadata` directory contains JSON metadata files named "metadata_<ID>" for each image. Each JSON file includes a list of Python dictionaries, one for each cell object visible in the image. Each dictionary contain the following keys:

- **ID**: Unique identifier for the object
- **ID_Type**: ID of cell type. Possible values: 1 for epithelial cells, 2 for goblet cells, 3 for plasma cells, 4 for lymphocytes, 5 for eosinophils, and 6 for fibroblasts.
- **Type**: Short name of cell type. Possible values: "EPI" for epithelial cells, "LYM" for lymphocytes, "PLA" for plasma cells, "FIB" for fibroblasts, "EOS" for eosinophils, and "GOB" for goblet cells.
- **Cellpart**: Specific part of the cell, either "Nucleus", "Cytoplasm" or "Goblet"
- **Cellname**: Unique name of that object
- **Staining_color**: RGBA color used for staining
- **Staining_intensity**: Intensity of the staining, between 0 and 100
- **Location**: 3D location of the object
- **Location_pixel**: Pixel coordinates of the object in the image

**Parameters**

The `parameters` directory contains JSON files named "parameters_<ID>", which detail the parameters used to generate each image. Each JSON file is a Python dictionary with all the parameter values necessary to reproduce the scene. The detailed parameters are:

- **seed**: The random integer seed used for reproducibility.
- **gpu_device**: The GPU device ID to be used for rendering.
- **gpu**: Boolean indicating whether to use GPU acceleration.
- **output_dir**: The directory where rendered images will be saved.
- **start_idx**: The starting index for naming rendered images.
- **n_samples**: The number of tissue samples to generate.
- **tissue_thickness**: The thickness of the tissue section.
- **tissue_size**: The size of the tissue section.

- **tissue_color**: The color of the tissue staining in RGBA format.
- **tissue_location**: The location of the tissue sample in 3D space.
- **tissue_padding**: The padding around the tissue sample. Is used to reduce the complexity of the whole macrostructure model to a smaller section.
- **tissue_rips**: The number of occuring tissue ripping instances.
- **tissue_rips_std**: The standard deviation of occuring tissue ripping instances.
- **stroma_intensity**: The intensity of stroma tissue staining.
- **noise_seed_shift**: The shift in noise seed.
- **stroma_density**: The density of stroma cells.
- **ratios**: The ratios of different cell types.
- **nuclei_intensity**: The intensity of cell nuclei staining.
- **mix_factor**: The shape mixing factor for plasma cell nuclei. 0 for pure plasma cell type shape, 1 for pure lymphocyte shape
- **epi_rescaling**: The rescaling factor for epithelial cells.
- **mix_cyto**: The shape mixing factor for plasma cytoplasm. 0 for pure plasma cell type shape, 1 for pure lymphocyte shape

**Variations**

The directory `variations` contains subdirectories named `<variation_type>`, each corresponding to specific aspects of the images which have been intentionally manipulated, while leaving other elements unchanged. Within each `variations_type` subdirectory, there are additional `<variation_type>_<img_noise>` subdirectories.

These subdirectories contain a subset of previously described `image`, `mask`, `parameters`, and `metadata` subdirectories, where the targeted aspect's parameter has been adjusted to the value `<img_noise>`. For instance, if the manipulation pertains to staining the cell nuclei, the parameter that is modified is **nuclei_intensity**. The detailed explanation of the parameters available for manipulation is provided in the Subsection F.2.

It is important to note that the structure of the `mask` directory presented here exclusively contains the `<task>_noise_0` directory. This directory specifically holds the original, unmodified masks related to the task influenced by the chosen `<variation_type>`.