# OpenReview forum: "Arctique: An artificial histopathological dataset unifying realism and controllability for uncertainty quantification"
_NeurIPS.cc/2024/Datasets_and_Benchmarks_Track — NeurIPS 2024 Track Datasets and Benchmarks Poster_

### Official Review · Reviewer_9out · 2024-06-30

**Rating:** 7
**Confidence:** 3
**Clarity:** Yes, the paper is well-written and or…

**Review:**

Pros:
- See strengths below.

Cons:
- The paper focuses on uncertainty quantification in image segmentation, while the title suggests a broader scope of uncertainty quantification. The title may appear overly general given the specific focus of the work.
- The dataset is derived from histopathological colon images and primarily targets cell segmentation tasks. This specialization could introduce biases when evaluating UQ algorithms, potentially limiting its applicability to other types of image analysis tasks.

**Strengths:**

- The benchmark dataset closely resembles real-world data and includes ground-truth annotations. Its realism is evaluated both qualitatively and quantitatively, making it a valuable resource for evaluating and developing UQ techniques.
- The dataset generation code is publicly available, allowing users to adapt it to their specific requirements.
- According to the description, the benchmark is versatile and can evaluate both 2D and 3D models.

**Additional Feedback:**

NA

**Correctness:**

The dataset is constructed soundly, and the evaluation methods are executed correctly.

**Documentation:**

The documentation is sufficient.

**Ethics:**

No ethical concerns

**Limitations:**

Yes

**Opportunities For Improvement:**

- Consider narrowing the scope of the topic.
- Address formatting inconsistencies, particularly in the citation of figures (some use "Fig.", some use "Figure").
- Clarify the meaning of "all three tasks" on line 210. Is it possibly a typo meant to say "all three algorithms" (TTA, dropout, ensemble)?

**Relation To Prior Work:**

Yes

**Summary And Contributions:**

Historically, uncertainty quantification (UQ) methods have been tested on simplistic toy datasets or on complex real-world datasets lacking ground-truth annotations. To address this gap, Arctique, a realistic and fully controllable dataset modeling histopathological colon images, is introduced to benchmark and advance UQ techniques.

---

> ### Author Rebuttal · Authors · 2024-08-16
>
> **Rebuttal Q&A**
>
> Thank you for your detailed and highly constructive review. We are pleased that you recognized that the realism and adaptability of our benchmark makes it a valuable resource for UQ method development and that you see the future potential for not only 2D but 3D models as well. We appreciate your feedback and have addressed your remarks individually below. If you have any further comments or suggestions, please let us know. We are more than happy to incorporate them.
>
>
> **(1) The paper focuses on uncertainty quantification in image segmentation, while the title suggests a broader scope of uncertainty quantification. The title may appear overly general given the specific focus of the work.**
>
> We want to thank the reviewer for pointing this out. Our experiments do focus on uncertainty quantification in image segmentation. We chose to focus on UQ methods within the segmentation context because accurate pixel-wise annotations are particularly hard to come by, which emphasizes the potential of our dataset. However, respective dense segmentation labels as well as procedural metadata allow to extract image level labels (as used in image classification or image-level regression) as well, and thus Arctique’s use is not inherently limited to image segmentation problems.
>
> Based on your feedback, we now conducted a proof-of-concept experiment: From the images and segmentation masks we extracted binary labels indicating the presence/absence of a given celltype. We then used the Arctique framework to alter the appearance of this cell and studied how this modification affects classification performance and uncertainty scores (see attached .pdf). This experiment could be extended and inserted as an ablation study in the Appendix of the paper. Would you deem this sufficient to stick to the general title of the manuscript?
>
> Note, we do of course still exclusively focus on image-based problems – The term “histopathological data” uniquely refers to image data, i.e. this scope is already captured in our title – However this meaning of “histopathological data” may not be clear to the general audience, so we could e.g. include“ image” before “dataset” in the title. What do you think? We highly appreciate your further feedback.
>
>
> **(2) The dataset is derived from histopathological colon images and primarily targets cell segmentation tasks. This specialization could introduce biases when evaluating UQ algorithms, potentially limiting its applicability to other types of image analysis tasks.**
>
> To address your concern, we will weave the following considerations into the Discussion section of our manuscript:
> We fully agree with the general concern that generalization to novel tasks for any method exclusively evaluated on domain-specific data/tasks remains unclear. Nonetheless, evaluating on a specific domain can still provide valuable insights, in particular if the test data comprises common features also present in other domains e.g. missing or incorrect labels, overlapping object instances, unknown objects or blur /out-of-focus in the case of Arctique.
>
> Furthermore, histopathology data is an attractive domain for uncertainty quantification as it is widely used for medical diagnosis and thus constitutes a prime example of a safety critical domain requiring reliable uncertainty quantification, boosted by the fact that the field is plagued by a lack of complete and accurate annotations.
>
> **(3) Address formatting inconsistencies, particularly in the citation of figures (some use "Fig.", some use "Figure").**
>
> Thanks for pointing this out, we revised our manuscript again to get rid of any formatting inconsistencies.
>
> **(4) Clarify the meaning of "all three tasks" on line 210. Is it possibly a typo meant to say "all three algorithms" (TTA, dropout, ensemble)?**
>
> Thank you for spotting this; we fixed it (we meant “both tasks”, i.e. the fg/bg and semantic segmentation).

---

> > ### Comment · Reviewer_9out · 2024-08-30
> >
> > Thanks to the authors for replying to my comments.

---

### Official Review · Reviewer_EfH6 · 2024-07-15
**Well-written, very relevant to the domain and well-evaluated**

**Rating:** 9
**Confidence:** 5
**Correctness:** Technically sound.
**Clarity:** Clearly written and encourages reader…

**Review:**

The manuscript is well-written and studies a real common problem in the field. Accurately studying uncertainty is very-much an ongoing research problem and a standardised dataset would definitely advance the understanding of uncertainty quantification methods. I believe this paper would b of great interest to the community.

**Strengths:**

- Clear presentation
- evaluation strategy very clear and insightful (zero-shot generalisation is a good mechanism to evaluate realism)
- data generation methods clear and technically sound

**Additional Feedback:**

I think this work studies an important problem in the field of medical imaging and would generate significant interest in the within the research community.

**Documentation:**

Adequate.

**Limitations:**

Adequately addressed.

**Opportunities For Improvement:**

- The uncertainty quantification methods evaluated are commonly used in practice but are a bit outdated. Recent quality assessment and active learning literature provide more recent better-performing methods for uncertainty quantification e.g.,:
- https://www.sciencedirect.com/science/article/pii/S1361841522000780
- https://link.springer.com/article/10.1007/s10994-021-06003-9
- https://www.sciencedirect.com/science/article/pii/S1361841524001063
- https://arxiv.org/abs/1708.02383
- https://arxiv.org/abs/1702.06559
These methods can be used as baselines or as another field for evaluation using this dataset. A brief discussion could be added on these methods and the potential of this data set in the field of active learning or quality assessment. These would be good to see as comparisons, however, not essential for publication of this work as the commonly used methods have already been evaluated.

**Relation To Prior Work:**

Adequately addressed for this work, although could be expanded as an improvement (see opportunites ofr improvement).

**Summary And Contributions:**

The authors generate a synthetic dataset for histopathology imaging tasks to evaluate uncertainty quantification methods. The authors argue that real uncertainty in real medical imaging datasets is prevalent in terms of label and imaging noise but its quantification remains challenging due to observer variability etc. This is why a synthetic dataset is valuable. Authors first investigate realism of the data by doing a zero-shot generalisation from synthetic to real samples and then quantify uncertainty using methods used in common practice.

---

> ### Author Rebuttal · Authors · 2024-08-16
>
> **Rebuttal Q&A**
>
> Many thanks for your highly constructive feedback. We were particularly happy to read that you perceived our paper to be of great interest to the community. Furthermore, we are grateful for your detailed literature recommendations in the realm of Active Learning and helpful suggestions for further improving our manuscript. We have made some changes to our Discussion section to incorporate your suggestions. If there are any further aspects you would like us to address please don’t hesitate to reach out.
>
>
> **(1) uncertainty quantification methods evaluated are commonly used in practice but are a bit outdated**
>
> Precisely as you observed, when selecting the uncertainty quantification (UQ) methods to test on our dataset, we prioritized widely employed methods over latest innovations. Given that the main contribution of our work is the preparation and provision of a dataset designed to facilitate the controlled study of uncertainty in image segmentation, we deemed it most valuable for practitioners to observe how tried-and-true methods behave in this new testbed.
>
> **(2) A brief discussion could be added on these methods and the potential of this data set in the field of active learning or quality assessment.**
>
> We greatly appreciate your suggestion to expand our Discussion to stress the relevance of our dataset for the highly active field of AL. We will incorporate your recommendations into the Discussion section of the paper with the following text:
>
> *“While our main goal in this publication is to introduce the Arctique dataset and illustrate its utility for evaluating UQ methods, an important follow-up would be to expand the range of studied UQ methods and conduct extensive benchmarking. Active Learning (AL) represents a particularly compelling area of study due to its long-standing use of uncertainty as a sampling strategy. Recent research has highlighted the significance of uncertainty disentanglement in AL, showing that requesting labels for samples with high epistemic uncertainty enhances performance, whereas focusing on samples with high aleatoric uncertainty can be detrimental (Czolbe et al. 2021, Nguyen et al., 2021). We believe that Arctique's ability to control the level of uncertainty in the data can support the evaluation of AL sampling strategies, the integration of uncertainty principles into learning optimization, and ultimately, the feasibility of domain adaptation (Fang et al. 2017, Woodward and Finn 2017, Saeed et al. 2021, 2024). “*

---

> > ### Comment · Reviewer_EfH6 · 2024-08-28
> >
> > The authors have sufficiently addressed my concerns, however, I am still unable to see an updated PDF. I would encourage the authors to upload this as soon as possible so the new changes can be viewed in context of the entire manuscript. I would like like to keep my original positive rating.

---

> > > ### Author Rebuttal · Authors · 2024-08-28
> > >
> > > Many thanks for your response. We are delighted to hear that you find your concerns sufficiently addressed.
> > >
> > > Please note that unfortunately the NeurIPS guidelines explicitly prohibit us from uploading an updated manuscript during the rebuttal phase (see "NeurIPS 2024 FAQ for Authors" subsection "Reviewing/Discussion process": “Can we upload a revision of our paper during the rebuttal/discussion period? -- No revisions are allowed until the camera-ready stage.”)  What we can do to address your request is to specify the precise line numbers where we will add the paragraph about active learning (italicized in our rebuttal): It will be put between lines 266 and 267 of the Discussion in the current version of the manuscript.
> > >
> > > We sincerely hope that this provides sufficient context regarding the changes to our manuscript; In case you have any further questions please do not hesitate to contact us.

---

### Official Review · Reviewer_1Q23 · 2024-07-16
**An artificial histopathology dataset for quantifying uncertainty**

**Rating:** 4
**Confidence:** 3
**Correctness:** The dataset and benchmark are constru…
**Clarity:** The paper is generally well written.

**Review:**

Soundness: 2 fair Presentation: 2 fair Contribution: 2 fair

**Strengths:**

The paper is well-written and easy to follow.
The proposed framework is technically sound.
The experiments are comprehensive.

**Additional Feedback:**

Please address my issues above.

**Documentation:**

The authors claim that they will publish the dataset while I didn't found the dataset on the provided website.

**Limitations:**

The authors provide detailed discussion for the limitations. Have the authors considered releasing the full pretrain data along with the pretrained model? Computer vision and natural langauge processing communities have grown so much because of the open-source pretrained data like ImageNet or C4 datasets. Otherwise this should be considered as one of the limitations of the work.

**Opportunities For Improvement:**

Great results, but lacks novelty, analysis, and extensive ablations.
There is no visualization comparison between different methods.

**Relation To Prior Work:**

The related works are clear.

**Summary And Contributions:**

This paper introduces a procedurally generated dataset modeled after histopathological colon images to efficiently study the performance of several commonly used UQ methods by independently controlling for uncertainty in images and labels.

---

> ### Author Rebuttal · Authors · 2024-08-16
>
> **Rebuttal Q&A**
>
> Thank you for taking the time to provide feedback on our work. We've carefully considered your comments and have done our best to address them. However, there are a few points where we would greatly appreciate further clarification to ensure we fully understand your concerns and can respond appropriately.
>
>
> **(1) Lacks novelty.**
>
> We respectfully disagree. To date, uncertainty quantification methods have been tested either on simplistic toy datasets or on complex real-world datasets lacking ground-truth annotations. Our proposed dataset fills this gap by offering a unique combination of real-world image complexity and full controllability. This novel dual capability makes the dataset exceptionally suitable for evaluating and developing uncertainty quantification methods.
>
> **(2) Lacks analysis and extensive ablations.**
>
> We would appreciate more specific guidance on which aspects of the analysis or additional ablations the reviewer believes are lacking. This would help us address the mentioned concerns more effectively in our revisions.
>
> **(3) Have the authors considered releasing the full pretrain data along with the pretrained model? / The authors claim that they will publish the dataset while I didn't find the dataset on the provided website.**
>
> Yes, we released the complete dataset. We included a link to the dataset in the supplementary material under “B Dataset Access”. If you are facing any issues accessing the dataset, please let us know and we’ll be happy to assist further. In addition, we will release the accompanying codebase for dataset generation upon publication. Our primary aim is to provide a public dataset specifically designed for the evaluation of uncertainty quantification methods. To illustrate the dataset's potential, we have conducted experiments based on standard models we (pre-)trained on our dataset. We will release all code for experiments and evaluation as well as our respective trained models upon publication.
>
> **(4) There is no visualization comparison between different methods.**
>
> We believe that the reviewer might be referring to visual comparisons of the uncertainty quantification methods. In Figure 3 and 5 of the paper, we provide visual comparisons of measures for (predictive, epistemic and aleatoric) uncertainty respectively based on the four studied approaches:  Monte-Carlo Dropout, Test Time Augmentation, Deep Ensembles, and Maximum Softmax Response. Furthermore, in Figure 4, we compare the resulting error-maps with the estimated predictive uncertainty for Monte-Carlo Dropout and Test Time Augmentation. The results of these visualizations are then discussed in section 4 of the paper. Finally, in Figures 8, 9 and 10 in the Appendix we compare the impact of three different aggregation strategies on the uncertainty scores.
> We are happy to provide more visualizations to further illustrate the results of our study. However, we would appreciate further clarification of what kinds of visualization the Reviewer has in mind so that we can address the concern more effectively.

---

### Author Rebuttal · Authors · 2024-08-16

We sincerely appreciate the positive and encouraging feedback from the reviewers. In our paper we presented a novel, procedurally generated, synthetic dataset embodying controllability while maintaining real-world complexity. We demonstrated how it can be used to evaluate uncertainty quantification methods for image segmentation by explicitly controlling the amount of epistemic and aleatoric uncertainty present in the data. We are delighted that the value of our proposed dataset in evaluating and developing uncertainty quantification techniques was highlighted (EfH6, 9out) and grateful for the acknowledgment of our paper's relevance to a common problem in the field (EfH6). Furthermore, we are pleased that reviewers noted the comprehensiveness (1Q23) and correctness (9out) of our experiments, and that the paper was recognized for being well-written (1Q23, EfH6, 9out).

We address all reviewers' remarks in individual replies. We hope that we answered them sufficiently and clarified any open questions. If you have any further comments, follow-ups or suggestions, please let us know. We highly appreciate it.

---

### Decision · Program_Chairs · 2024-09-26

**Decision:**

Accept (Poster)

**Comment:**

The controllability and complexity of the dataset presented in this paper makes it a valualbe contribution, filling a gap between real-world datasets and simplistic synthetic datasets. The reviewers appreciated the technical soundness, comprehensive experiments, and the potential impact of Arctique on the UQ and medical imaging communities. Although one reviewer noted a lack of novelty, I feel that the authors adequately addressed these concerns. Overall, the dataset is a strong contribution that will benefit the research community and support further advancements in UQ techniques. Based on the positive reviews and the authors' clarifications, I recommend acceptance.